# DREAMLAND: CONTROLLABLE WORLD CREATION WITH SIMULATOR AND GENERATIVE MODELS

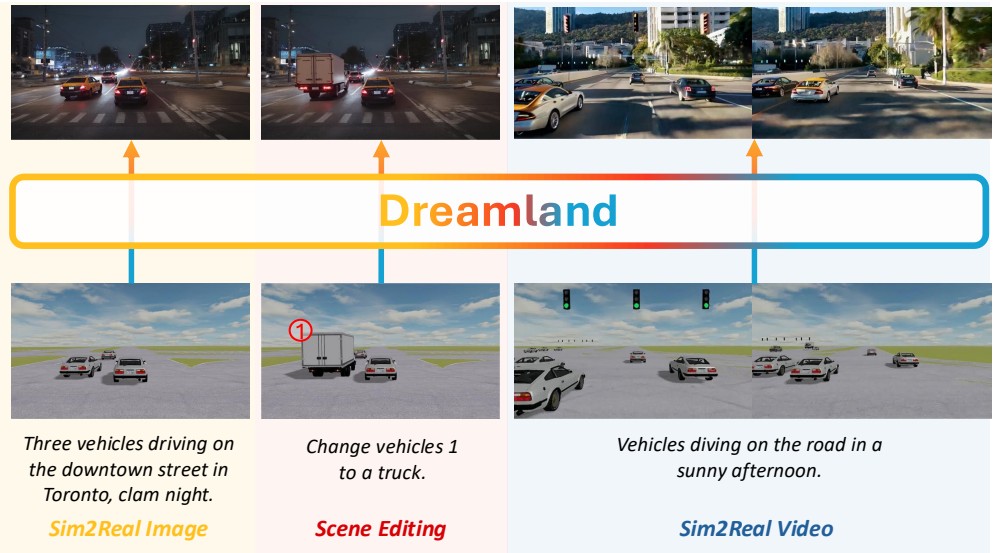

Figure 1: **Controllable world creation with Dreamland**. It combines a simulator for physically grounded scene generation and a large-scale pretrained generative model for creating a realistic visual world following user-provided text prompts.

## ABSTRACT

Large-scale video generative models can synthesize diverse and realistic visual content for dynamic world creation, but they often lack element-wise controllability, hindering their use in editing scenes and training embodied AI agents. We propose Dreamland, a hybrid world generation framework that combines the granular control of a physics-based simulator with the photorealistic content output of large-scale, pretrained generative models. In particular, we design a layered world abstraction that encodes both pixel-level and object-level semantics and geometry as an intermediate representation to bridge the simulator and the generative model. This approach enhances controllability, minimizes adaptation cost through early alignment with real-world distributions, and supports the off-the-shelf use of existing and future pretrained generative models. We further construct a D3Sim dataset to facilitate the training and evaluation of hybrid generation pipelines. Experiments demonstrate that Dreamland outperforms existing baselines with $50.8\%$ improved image quality, $17.9\%$ stronger controllability, and has great potential to enhance embodied agent training. Code and data will be made available.

## 1 INTRODUCTION

Large-scale pre-training with scalable model architectures has significantly advanced generative modeling in recent years. This progress has led to a surge of foundation models across various modalities, including language (Touvron et al., 2023; Grattafiori et al., 2024a; Mesnard et al., 2024; Zhu et al., 2023; Liu et al., 2023b;a), image (Rombach et al., 2022; Podell et al., 2024; Esser et al., 2024; Nichol et al., 2022; Ramesh et al., 2021; 2022), and video generation (OpenAI, 2025; Zheng et al., 2024b; Lin et al., 2024a; Ma et al., 2024). These foundation models capture structured

information about entities, their interactions, and temporal dynamics—often referred to as the "world knowledge". As such, they hold great promise for providing synthetic data to train embodied AI agents beyond just generating visual content. By providing interactive feedback, they can potentially replace human supervision or physical environments, enabling efficient and scalable agent learning.

Despite the capabilities of current world models being exciting, they often lack the fine-grained control required for agent learning, e.g., autonomous driving. For instance, training autonomous vehicles requires simulating complex scenarios. These include specific vehicle maneuvers like lane changes in dense traffic or responses to stop signs. Current generative world models often struggle to offer object-level controllability, thereby constraining their efficacy in situations where precise scene layout and object configuration are critical.

Recognizing these challenges, a hybrid approach that combines physical simulators and data-driven generative models has emerged as a promising solution. In such pipelines, the simulators first provide the accurate physical and spatial information, such as vehicle dynamics, traffic rules, and environmental function zones. Then, image or video generative models synthesize realistic visual content following these conditions. Despite these improved pipelines, current hybrid methods struggle to strike a balance between simulator fidelity and generative freedom. Some methods directly render scenes without any hallucination (Yu et al., 2024; Li et al., 2025), which necessitates high-fidelity simulators and compromises their generalizability and scalability for creating diverse worlds. On the other hand, other work (Zhou et al., 2024b) fully utilizes generative models for scene re-rendering without stringent constraints, which consequently sacrifices fine-grained control over the simulator.

To balance the controllability from simulators and the creative freedom of generative models with rich visual details in a synergetic way, we present Dreamland, bridging simulator and generative models through Layered World Abstraction (**LWA**). LWA records structured scene information that contains pixel-level details, such as depth and RGB values, as well as object-level information, including object categories. Instead of directly generating the scene from the initial simulator layered world abstraction (**Sim-LWA**), Dreamland first augments it to real-world layered world abstraction (**Real-LWA**) that better aligns with real-world visual distributions. During inference, Dreamland follows user instructions to divide the scene into preserved and editable regions. Editing models could help diversify the scene structure within the editable region while keeping the preserved region untouched. Therefore, by strictly following conditions in Real-LWA during visual re-rendering, the final generated scenes can achieve both high-quality appearance and precise object control that adheres to the simulator and user instructions.

Dreamland design offers several strengths. First, our pipeline enhances controllability and flexibility over the hybrid approach, outperforming the previous state-of-the-art (Zhou et al., 2024b) by 52.3% and 17.9% in image quality and controllability. Notably, Dreamland also demonstrates benefits in downstream agent training, boosting visual question answering performance on the real-world test set by 3.9 absolute accuracy. Second, since the Real-LWA is already aligned with real-world distributions, it can be seamlessly integrated with future, more potent conditional-generation models without introducing significant adaptation costs. Finally, Dreamland pipeline is generalizable and scalable, thus unlocking various applications based on simulators, including video generation and scene editing, as illustrated in Figure 1. We also construct a large-scale dataset for training and evaluating such hybrid pipelines.

We summarize our key contributions as follows: (1) We propose layered world abstraction (**LWA**), a novel and flexible design that connects a physics-based simulator and generative models to a hybrid generation pipeline **Dreamland**. (2) Our LWA design unlocks flexible applications and ensures granular controllability for downstream tasks. It demonstrates superior results in the scene generation task, exceeding previous state-of-the-art by 52.8% image quality and 17.9% controllability, and improves the adaptation of embodied agents to the real world. (3) We construct a dataset called **D3Sim** (**D**iverse **D**riving Scenario in Real Worl**D** and **Sim**ulation) for training and benchmarking hybrid generation pipelines that combine simulators and generative models.

## 2 RELATED WORK

**Controllable Visual Generative Models**. Large-scale foundational generative models have been developed rapidly in recent years for image generation (Rombach et al., 2022; Podell et al., 2024; Esser et al., 2024; Nichol et al., 2022; Ramesh et al., 2021; 2022), video generation (OpenAI, 2025;

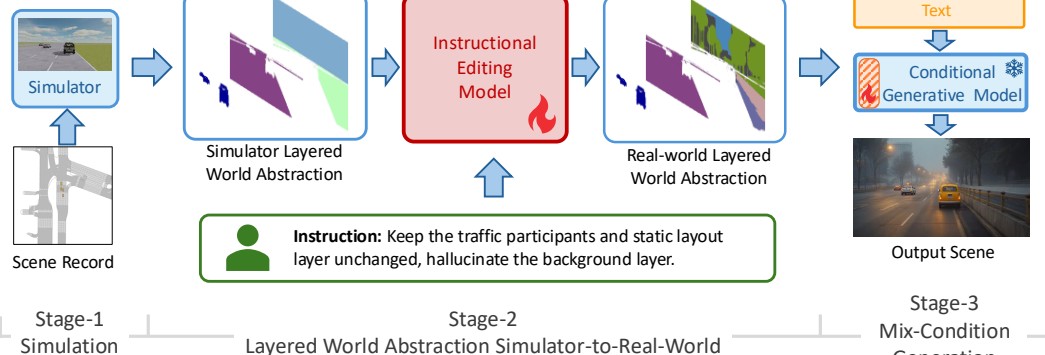

Figure 2: **Illustration of the Dreamland**. It is a three-stage pipeline including: (1) Simulation, (2) Layered World Abstraction Simulator-to-Real-World, (3) Mix-Condition Generation.

Zheng et al., 2024b; Lin et al., 2024a; Ma et al., 2024), and multimodal generation (Zhou et al., 2024a; Xie et al., 2024; Shi et al., 2024; Sun et al., 2023; Wang et al., 2024; Mo et al., 2025). While those foundation models usually rely solely on text prompts, various approaches have been proposed to add additional controllability to the generation process, including structure control (Zhang et al., 2023; Li et al., 2023c; Mou et al., 2024; Ju et al., 2023; Mo et al., 2024; Tumanyan et al., 2023; Lin et al., 2024b), subject control (Ruiz et al., 2023; Ye et al., 2023; Avrahami et al., 2023; Mokady et al., 2023), and adding image editing capability (Brooks et al., 2023; Mao et al., 2025). Meanwhile, another line of work addresses the more challenging multi-condition generation problem. Specifically, UniControl (Zhao et al., 2023a) and UniControlNet (Zhao et al., 2023b) propose unified adapters to process various conditioning signals. Distinguished from the above works that could obtain high-quality paired data with pre-trained detection models, the simulator and real-world paired data are expensive to annotate. Therefore, Dreamland focuses on adapting from existing conditional generation models with low cost while preserving their rich world knowledge.

**Generative Model for Autonomous Driving**. Generative models have largely advanced autonomous driving research in recent years. One line of work (Swerdlow et al., 2024a; Yang et al., 2023; Gao et al., 2024b;a; Wang et al., 2023; Li et al., 2023b) trains generative models, including GANs and Diffusion Models, to generate multi-view images from a driving scene graph, bird-eye-view, or HD maps. Specifically, Panacea (Wen et al., 2024), InfiniteCube (Lu et al., 2024), MagicDrive (Gao et al., 2024b), and MagicDrive-V2 (Gao et al., 2024a) build on pre-trained video diffusion models to add temporal consistency into the generated scene. Another line of work (Zheng et al., 2024a; Gao et al., 2024c; Hu et al., 2023; Kim et al., 2021) takes historical information as the condition signal to generate future driving scenes, but they often lack control over the generated scenes. More recently, Cosmos-Drive-Dreams (Ren et al., 2025) extends the world foundation models to driving scenarios. Dreamland differs from previous literature in its hybrid approach, which connects the generative models with simulators with our novel LWA, thus enabling fine-grained control over each object and region, such as specifying their individual paths, actions, and interactions.

**Synthetic Data for Embodied AI Training**. The significant visual domain gap between simulators and the real world data has historically presented a challenge in training agents in simulation. Due to recent advances in generative and reconstruction techniques, this problem has been partially addressed by integrating simulators, vision generative models, and 3D reconstruction methods. For example, Vid2Sim (Xie et al., 2025) and VR-Robo (Zhu et al., 2025) employ Gaussian splatting techniques to convert a real scene into an interactive digital twin, allowing trained agents to zero-shot deploy in the real world. On the other hand, Lucidsim (Yu et al., 2024) trains a robot dog to achieve great adaptability using a simulator with a depth-conditioned diffusion model. Compared to previous literature, Dreamland ensures granular controllability while applying to various generative models.

## 3 METHOD

Dreamland enables controllable driving scene creation via three key stages shown in Figure 2: (i) *Stage-1 Simulation*: scene construction with physics-based simulator. (ii) *Stage-2 LWA-Sim2Real*: transferring the Sim-LWA from simulation to Real-LWA with an instructional editing model and user instructions. (iii) *Stage-3 Mixed-Condition Generation*: rendering an aesthetic and realistic scene with a large-scale pretrained image or video generation model. The pipeline begins with the simulator rendering a scene according to the scene record, where the agents' motion is derived from either

trajectory replay or a trained policy. The Sim-LWA captured from the simulator is then refined and rendered into realistic frames. We first introduce the layered world abstraction in §3.1. We then talk about the design of Stage-2 in §3.2, and Stage-3 in §3.3. Lastly, we describe our training scheme and implementation details in §3.4.

## 3.1 Layered World Abstraction

We define Layered World Abstraction (LWA) as an intermediate representation that aligns simulators and generative models in pixel space. It enables fine-grained control over the generation process. LWA composes a scene from multiple world layers, where each layer corresponds to different classes of objects or regions. The representation is structured as

$$\mathcal{W} = \bigcup_{i=0}^{N} \mathcal{L}_i \odot \mathcal{V}_i, \tag{1}$$

where $\mathcal{L}_i \in \mathbb{R}^{H \times W \times D}$ denotes the $i^{\text{th}}$ world layers, and $\mathcal{V}_i \in \mathbb{R}^{H \times W}$ is the corresponding visibility mask. Each world layer encodes both pixel-level and object-level geometry and semantics through $K$ modalities of condition. Thus, the $i^{\text{th}}$ layer is formulated as

$$\mathcal{L}_i = \left\{ \mathbf{c}_j \in \mathbb{R}^{H \times W \times C_j} \middle| j = 0, 1, \ldots, K; \sum_{j=0}^{K} C_j = D \right\}, \tag{2}$$

where $\mathbf{c}_j$ denotes the $j^{\text{th}}$ condition with $C_j$ channels. This flexible design enables customizable, pixel-level control over both the object of interest and the region of interest.

Given a driving frame rendered by the simulator following the scene record, we decompose it into Sim-LWA according to the pixel semantics. Sim-LWA contains accurate traffic participants and layout, which is physically grounded by the simulator and serves well for layers requiring precise control. However, due to the limited assets and lack of background details, those corresponding layers exhibit a disparity from real-world distribution, which hinders the realistic scene generation.

## 3.2 Stage-2: LWA-Sim2Real

The LWA-Sim2Real refinement process involves refining the Sim-LWA to real-world distributions. In the Dreamland pipeline, we compose our LWA from three world layers: a traffic participants layer $\mathcal{L}^d$ that contains dynamic objects (e.g., vehicles, pedestrians, cyclists), a map layout layer $\mathcal{L}^l$ that defines a static layout (e.g., roads, crosswalk, intersections), and a background layer $\mathcal{L}^b$ that covers static objects and background regions (e.g., buildings, vegetation, sky). The first two layers, $\mathcal{L}^d$ and $\mathcal{L}^l$, are derived from the simulator and preserved to provide precise control. In contrast, the last layer $\mathcal{L}^b$ is editable and refined by an editing model as described below.

Given preserved world layers and their masks from simulators $\{\mathcal{L}^d, \mathcal{L}^l\}, \{\mathcal{V}^d, \mathcal{V}^l\}$, we employ an instructional editing model $\epsilon_e$ to selectively refine layer $\mathcal{L}^b$ according to the provided text instruction $c$. This process can be formulated as,

$$\mathcal{L}^b = \epsilon_e \left( \hat{\mathcal{W}}, \mathcal{V}^b, c \right). \tag{3}$$

The $\hat{\mathcal{W}} = \left\{ \mathcal{L}^d \odot \mathcal{V}^d \cup \mathcal{L}^l \odot \mathcal{V}^l \right\}$ denotes the preserved LWA from the simulator, and the $\mathcal{V}^b = \Omega \setminus (\mathcal{V}^d \cup \mathcal{V}^l)$ is the editing mask given the pixel domain $\Omega$.

This process transfers the Sim-LWA to real-world distribution while preserving the grounding information from the simulator. It bridges the domain gap between the general simulators and generation models, alleviating the requirements for high-fidelity simulation rendering and minimizing the adaptation costs for generation models.

## 3.3 Stage-3: Mixed-Condition Generation

In this stage, we utilize a pre-trained conditional generative model to generate realistic views according to our refined Real-LWA. As the representation is already aligned with real-world distribution, minimal adaptation is required to employ a pre-trained model on it, thus preserving its world knowledge. For pre-trained conditional world models that cover the condition modalities in our LWA,

we split the channel dimension of LWA according to condition modalities and serve as the respective condition inputs.

However, some of the pre-trained conditional world models are trained on specific condition modalities that do not fully cover the modalities within our LWA. Therefore, we have two choices to further minimize the adaptation cost while preserving the pre-trained world knowledge: (1) Extracting only specific condition modalities, e.g., depth or segmentation maps, from LWA. (2) Adapting the pre-trained conditional world model to our LWA by updating a small number of parameters. We empirically found that the second approach yields better controllability, and thus adopted it in our design.

Given the Real-LWA, which contains $K$ conditions, we first encode them into the latent space with encoder $\mathcal{E}$, and concatenate them along the channel dimension. Then, we use a linear layer $\xi$ to project the concatenated condition latent into the same dimension as the noise latent $\mathbf{x}$ of the world model. Finally, we add the projected condition latent with the noise latent to incorporate structural guidance. The whole process can be formulated as

$$\mathbf{x}' = \mathbf{x} + \xi\left(\bigoplus_{j=0}^{K}\mathcal{E}(\mathbf{c}_j)\right). \tag{4}$$

The output noise latent $\mathbf{x}'$ with structural information serves as the input to the diffusion transformer blocks of the world model. Thus, we achieve mixed-condition generation by fine-tuning the projection layer $\xi$, which renders a realistic scene from the refined LWA.

### 3.4 Training Scheme and Implementation Details

We train our Dreamland pipeline in two steps. The first training step corresponds to the LWA-Sim2Real stage, where it enables the instructional editing model to learn Sim2Real refining of LWA. The second step is for the mixed-condition generation stage, which involves fine-tuning a conditional generation model to take structural control from the Real-LWA. We design our instructional editing model following ACE++ (Mao et al., 2025), training on our image dataset for 4K iterations with a batch size of 128. To align with the pre-trained editing model, we expand the world representation into a single image by vertically concatenating the condition maps. The loss for LWA Sim2Real transfer follows

$$L_{\text{Sim2Real}} = \mathbb{E}_{\mathcal{L}_0^b}\left[\|\mathcal{L}^b - \mathcal{L}_0^b\|_2^2\right], \tag{5}$$

where $\mathcal{L}^b$ is the refined layer defined in Eq. 3 and $\mathcal{L}_0^b$ denotes the corresponding world layer in real-world distribution. We employ Flux Depth (Black Forest Labs, 2024), an image generation model $\epsilon_\theta$ with structure control based on depth maps, for our second step and adapt it to our LWA using the loss formulated as

$$L_{\text{adapt}} = \mathbb{E}_{\mathbf{x}_0}\left[\|\epsilon_\theta(\mathbf{x}') - \mathbf{x}_0\|_2^2\right], \tag{6}$$

where the $\mathbf{x}'$ is from Eq. 4 and $\mathbf{x}_0$ is the latent of the realistic view. To balance the inference cost and visual quality, we set the default resolution (width) of our second- and third-stage models to 512 and 1024, respectively.

## 4 Data Curation

We curate a large-scale dataset called D3Sim (**D**iverse **D**riving Scenario in Real Worl**D** and **Sim**ulation) that contains diverse driving scenarios in the real world and the simulation. It provides realistic perspective views and high-quality condition data to facilitate the Sim2Real transfer. Previous digital twin driving datasets (e.g., DIVA-real (Zhou et al., 2024b)) comprise limited samples with the image mismatch problem, which is suboptimal for training our Stage-2 model. Thus, we precisely aligned our digital twin driving scenes for LWA-Sim2Real transfer. Based on the nuPlan dataset (Caesar et al., 2021), we obtain realistic conditions *(Real Conditions)* using pre-trained models. We then utilize ScenarioNet (Li et al., 2023a) to construct the corresponding digital twin simulation scene in the MetaDrive simulator (Li et al., 2022). By replaying the ego vehicle trajectory, we obtain conditions in the simulation domain *(Sim Conditions)*. We organize the constructed digital twins into two dataset variants for the Dreamland pipeline. Details of the data curation pipeline, as well as the video dataset used in the Dreamland-Video pipeline, are provided in Appendix Sec. B.

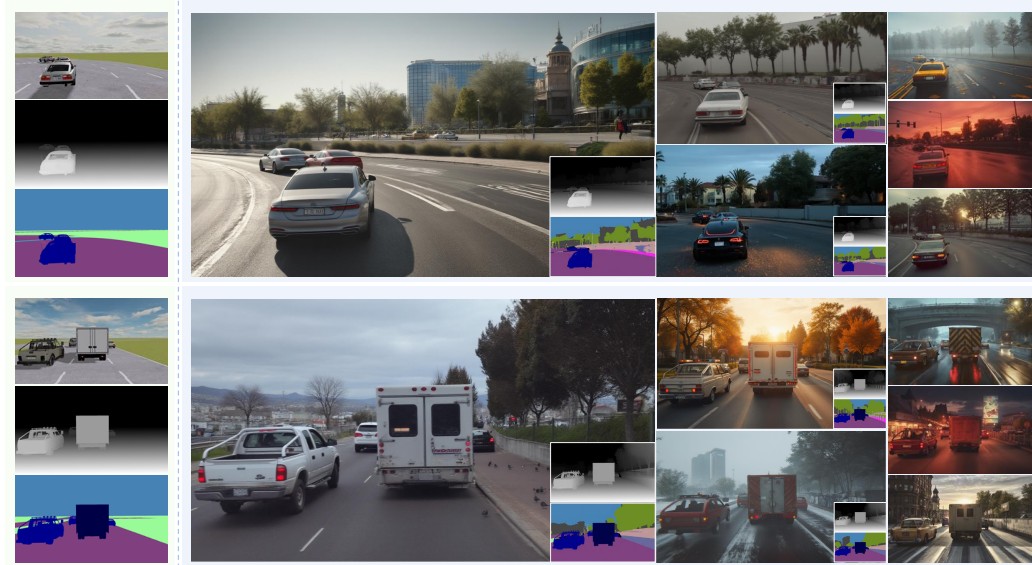

Figure 4: **Rendering diverse appearances aligns with the simulator's conditions**. Dreamland can simulate the driving scene and render it into high-quality scenes with generative models.

**Training Dataset**. This dataset contains digital twin driving scenarios that capture the real-world distribution, enabling the second-stage model to learn Sim2Real transfer. We construct the dataset with paired Sim and Real conditions, along with realistic perspective views. We split the conditions into three world layers according to the semantics and construct the LWA accordingly. This results in high-quality digital twin training data from approximately 1,800 scenarios with around 60,000 samples at a 2 Hz sample rate. We use this dataset to train our Dreamland pipeline.

**Validation Dataset**. The validation dataset is used to evaluate both the visual quality and controllability of the generation pipeline. While the digital twin scenarios provide realistic views and distribution of the real-world driving, the complex scene compositions, including partially occluded or overlapping objects, affect the accurate evaluation of controllability. Therefore, we derived this validation dataset from real-world scenarios by selectively modifying the traffic participants, eliminating ambiguous object placements, while keeping the scene layout unchanged. Diverse text prompts involving more than 30 cities, 20 weather conditions, different times of day, and various street types are randomly generated for each sample. This yields diverse and clean validation data from more than 600 scenarios with 16,000 samples at a 2 Hz sample rate, with object distribution shown in Figure 3.

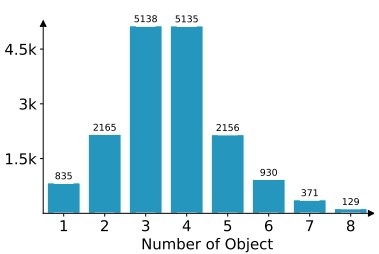

Figure 3: Object distribution of the validation dataset.

## 5 EXPERIMENTS

Extensive qualitative and quantitative results demonstrate the proposed pipeline's generation performance (Sec. 5.1). We also present experiments on flexible control using Dreamland, an extension to Dreamland-Video for continuous simulator re-rendering (Sec. 5.2), an ablation study of our key designs (Sec. 5.3), and an application to a downstream agent learning task (Sec. 5.4).

**Evaluation Metrics.** There are two aspects regarding the evaluation of the simulator-controlled world creation pipeline: the fidelity of the re-rendered scene to the original simulator output, and the overall visual quality of these visual re-renderings. We render 16K scenes from our D3Sim validation dataset and report the FID to DIVA-Real Dataset (Zhou et al., 2024b). We also evaluate depth si-RMSE and segmentation mIoU following previous works (Zhang et al., 2023; Agarwal et al., 2025).

### 5.1 SIMULATOR CONDITIONED GENERATION

**Qualitative Results.** As shown in Figure 4, Dreamland can re-render a scene into diverse high-quality scenes closely aligned with the spatial scene layout in the simulator but differ in weather, location,

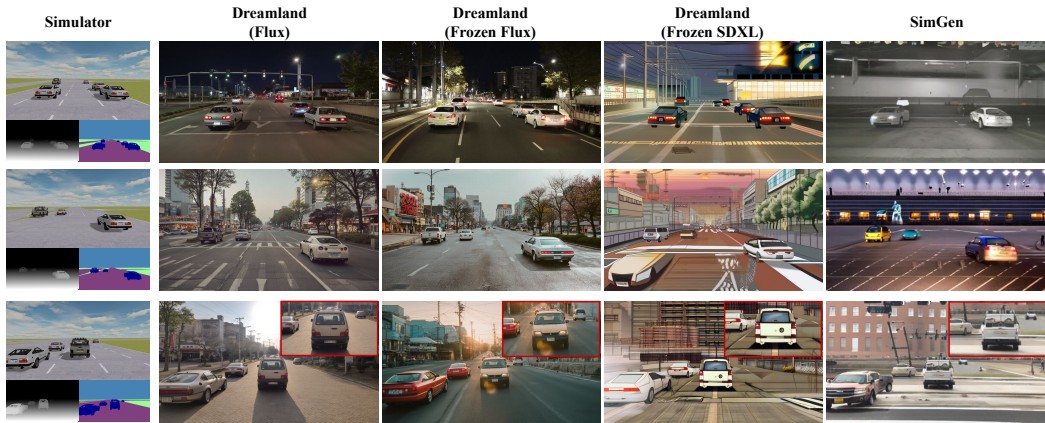

Figure 5: **Qualitative comparison** with baseline methods. Dreamland with fine-tuned Flux matches the visual fidelity of the frozen Flux model while improving adherence to simulator conditions.

and light condition following user-provided text prompts. Leveraging the robust capabilities of pre-trained Text-to-Image diffusion models, Dreamland is capable of synthesizing realistic driving scenes with a resolution of up to $1024 \times 576$ pixels.

**Baselines.** Our main baselines include generative models for autonomous driving that take scene layout conditions. SimGen (Zhou et al., 2024b) first obtains front-view observation from the MetaDrive simulator (Li et al., 2022), then trains cascade diffusion models to render high-quality images. Alongside our main approach, we create variations of Dreamland by extracting depth condition from our LWA and directly employing pretrained depth-conditioned diffusion models as third-stage models, notated as Dreamland (*Frozen {Model}*).

**Comparison with Baselines.**

Figure 5 and Table 1 compare our methods against several baselines. We observe that Dreamland vastly outperforms Sim-Gen with 52.3% lower FID and 17.9% better si-RMSE. Meanwhile, the Dreamland pipeline exhibits superior scalability: it benefits from plugging in progressively stronger pretrained models for Stage-3 (SDXL $\rightarrow$ SD3 $\rightarrow$ Flux), yielding marked gains in image quality and controllability. Notably, Dreamland fine-tuned on Flux

| Method | Image Quality | Controllability | |
|---|---|---|---|
| | FID ↓ | si-RMSE ↓ | mIoU ↑ |
| SimGen | 106.02 | 0.787 | 0.752 |
| **Dreamland** (*Frozen SDXL*) | 84.48 | 0.744 | 0.698 |
| **Dreamland** (*Frozen SD3*) | 63.74 | 0.673 | 0.747 |
| **Dreamland** (*Frozen Flux*) | 48.48 | 0.639 | 0.785 |
| **Dreamland** (*Flux*) | 50.58 | 0.646 | 0.791 |

Table 1: **Comparison on image quality and controllability**. Dreamland is strong and scalable.

matches the visual fidelity of the frozen Flux model while slightly improving adherence to simulator conditions, highlighting the world-knowledge preservation inherent in our Dreamland pipeline. We conduct additional evaluation on nuScenes and DIVA-Real in Appendix Sec. D.3.

## 5.2 EXTENSION OF DREAMLAND.

**Simulator Editing.** Figure 6 (a) shows that Dreamland supports a new application that edits a generated scene by adjusting the corresponding source scene. Building upon our current pipeline, we employ a pretrained Flux-fill model to remove the truck and add a speedy car on the road. By updating the LWA, we could provide additional editing mask support, such as a pretrained model without additional adaptation cost, showcasing the flexibility of our Dreamland pipeline.

**Safety-Critical Generation.** Figure 6 (b) shows that by designing the driving scenario with the simulator, Dreamland could generate safety-critical scenes that are dangerous to collect in the real world, revealing the great potential of applying Dreamland for agent learning.

**Extension to Additional Simulators.** With its simple and generalized pipeline design, our LWA naturally supports various simulator types. Figure 6 (c) demonstrates Dreamland's capability on the Isaac Sim simulator for a multiplication task. It could generate a realistic digital cousin by changing the background without modifying the target object's layout. We provide details on the

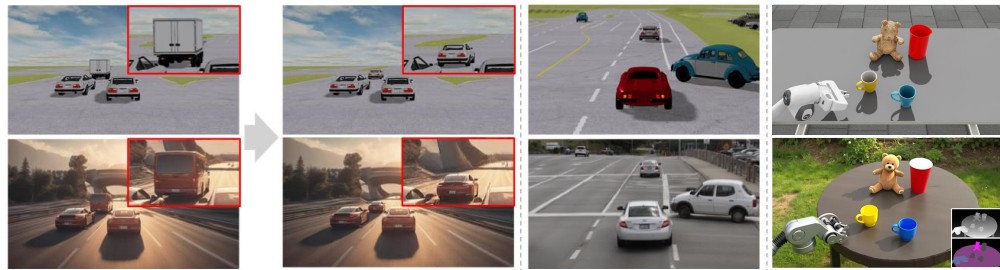

(a) Simulator–conditioned scene editing   (b) Safety-critical scenes   (c) Dreamland on Isaac Sim

Figure 6: **Extension of Dreamland**. Dreamland pipeline is generalized to various downstream tasks.

| | Image Quality | Pipeline Controllability | | Stage-2 Controllability | |
|---|---|---|---|---|---|
| Method | FID ↓ | si-RMSE ↓ | mIoU ↑ | si-RMSE ↓ | mIoU ↑ |
| Dreamland w/o editing | 63.21 | 0.713 | 0.658 | 0.709 | 0.649 |
| Dreamland | **50.58** | **0.647** | **0.791** | **0.672** | **0.950** |

Table 2: **Ablation Study on Stage-2 design choices.** Dreamland's Stage-2 design largely improved alignment to Stage-1's simulator, leading to better visual quality.

implementation of integrating Dreamland with Isaac Sim and MetaUrban simulators (Wu et al., 2025b) for robotic manipulation and mobility agent learning in Appendix Sec. D.3.

**Extension to Video Re-Rendering.** Benefit from the simple LWA design and low adaptation cost of Dreamland, it extends naturally to video generation without any specialized architectural changes. We provide the Figure 7 demonstrates the strong performance of our pipeline. By preserving the world knowledge embedded in these models, Dreamland-Video achieves the same level of controllability as Dreamland while consistently delivering high visual fidelity. We report the implementation details and additional quantitative results in Appendix Sec. D.6.

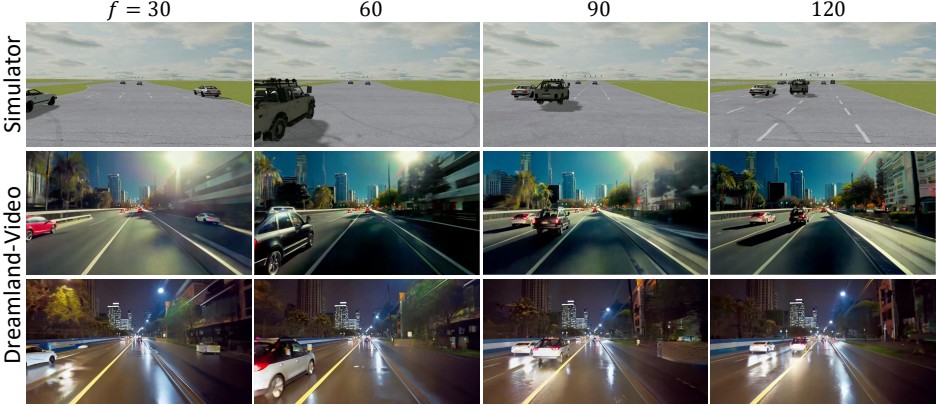

Figure 7: **Qualitative Results**. Our Dreamland-Video pipeline can re-render the simulator reference video into realistic and diverse video frames while maintaining the pretrained world knowledge.

## 5.3 ABLATION STUDY

We evaluate the effectiveness of our Stage-2 instructional editing model by comparing it against a baseline that directly generates the abstraction. Specifically, we finetune the Flux-Fill-Dev (Black Forest Labs, 2024) model via LoRA (Hu et al., 2022) for 4K iterations under two setting: (1) editing pipeline: our Real-LWA serves as targets, and we provide an editing mask to the model as input; (2) generation pipeline: segmentation and depth maps of the realistic views derived from pretrained networks are set as the learning target.

At inference time, we again provide an editing mask to indicate regions that should be preserved to our editing models, and we employ the same Stage-3 model to render these condition maps into realistic scenes. Across 16K generated samples on our validation set, qualitative results (Figure 8) reveal that, without the editing approach, the Stage-2 model struggles to produce condition maps with sharp, accurate object boundaries (e.g., vehicles and pedestrians) and loses the fine-grained control inherited from the simulator (e.g., specifying the exact height of a truck). We report the

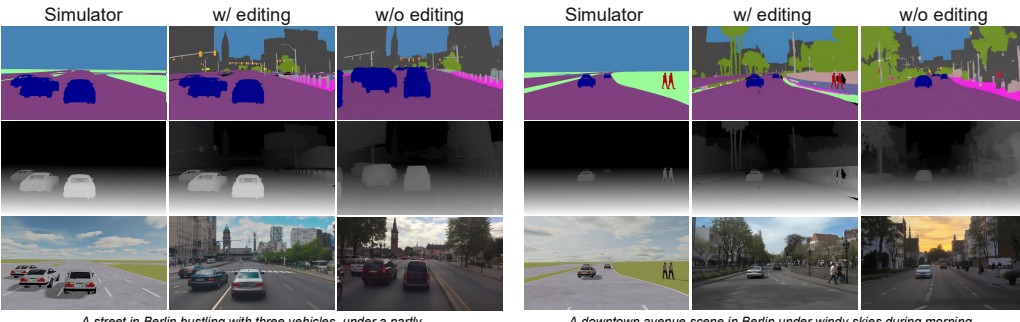

*A street in Berlin bustling with three vehicles, under a partly cloudy sky in the afternoon.*

*A downtown avenue scene in Berlin under windy skies during morning, featuring three vehicles and two pedestrians.*

Figure 8: **Qualitative Comparison** of Stage-2 model's design choices.

| Method | $AP_{Car}$ | $AP_{Truck}$ | $AP_{Bus}$ | $AP_{Ped}$ |
|---|---|---|---|---|
| Oracle | 47.6 | 21.2 | 45.3 | 33.9 |
| BEVGen | 24.7 | 9.1 | - | - |
| MagicDrive | 39.2 | 16.5 | 18.3 | **24.4** |
| SimGen | 41.0 | **19.6** | - | - |
| Dreamland | **42.4** | 14.1 | **40.8** | 23.6 |

Table 3: **3D object detection performance**. Oracle: a single-frame version of BEVFusion.

| Data | Overall | Synthetic | Real |
|---|---|---|---|
| S | 65.7% | 65.7% | 65.6% |
| S+D | 68.4% (+2.7%) | 67.3% (+1.6%) | 69.5% (+3.9%) |
| D | 74.5% (+8.8%) | 72.8% (+7.1%) | 76.3% (+10.7%) |

Table 4: **Improved Model Learning with Dreamland**. We use **S** and **D** to indicate the synthetic from MetaVQA and Dreamland generated data.

quantitative results in Table 2 and additionally evaluate the refined condition maps' alignment to the simulator condition within the preserved region. Stage-2's editing model, trained on our editing data, consistently outperformed the baseline, demonstrating the effectiveness of our pipeline design.

## 5.4 DOWNSTREAM TASKS

**3D Object Detection.** We conducted downstream evaluation on perception tasks. We use a pre-trained single-frame version of BEVFusion (Liang et al., 2022) for 3D object detection and report the AP of various classes in Table 3. Dreamland achieved the best performance in the Car and Bus classes and obtained competitive performance in other classes. This demonstrates the strong controllability of our pipeline and its potential for application in downstream autonomous driving tasks.

**Embodied Agent Training.** We further demonstrate that the output synthetic data of Dreamland as data augmentation can enable the downstream task of embodied agent training. As presented in MetaVQA (Wang et al., 2025), synthetic images benefit the learning of situational awareness for general-purpose Vision-Language Models (VLMs) through Visual-Question-Answering. To validate improved learning from enriched visual observations through Dreamland, we set up three LoRA (Hu et al., 2022) fine-tuning trials for InternVL2-8B (Chen et al., 2024). Generated from an identical set of top-down layouts, the first training set comprises only synthetic images rendered from MetaDrive (Li et al., 2022). In comparison, half of the second training set and all of the third training set are rendered via Dreamland. Question-answer pairs are generated for corresponding image sets, and the model is fine-tuned on three equally sized VQA sets and tested on a curated test set. Table 4 shows that the VLM model improves the test set especially on real-image-VQAs, and replacing the simulator-rendered observation with Dreamland's generation further boosts the test performance. These improvements showcase Dreamland's applicability in complex downstream tasks. Further experiment details will be in the supplementary materials.

## 6 CONCLUSION

We present Dreamland, a hybrid generation framework that, via a novel layered world abstraction (LWA), bridges physics-based simulators with large-scale pre-trained generative models to balance precise control and rich visual realism. To facilitate Sim2Real transfer for such systems, we introduce D3Sim, a large-scale dataset of paired simulated and real-world driving scenarios. Furthermore, we show Dreamland's strong semantic and geometric control enhances the real-world adaptation of downstream embodied agents. We hope our LWA and pipeline design can shed light on Sim2Real generation and more broadly embodied agent learning with generative models.

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

APPENDIX

For sections and figures, we use numbers (*e.g*, Sec. 1) to refer to the main paper and capital letters (*e.g*, Sec. A) to refer to this appendix. We hope this document complements the main paper.

# A   USAGE OF LLM

We used the LLM to check the grammar and revise a few sentences in the introduction to improve the writing quality.

# B   D3SIM DATASET

## B.1   D3SIM DATASET CONSTRUCTION

**Data preparation**.

We construct our D3Sim dataset based on the nuPlan dataset (Caesar et al., 2021), which contains diverse real-world driving scenarios and more than 120 hours of sensor data. To obtain the driving data in the simulation domain, we reconstruct the digital twin of real-world driving scenarios in the MetaDrive simulator (Li et al., 2022) using ScenarioNet (Li et al., 2023a). This results in scene records corresponding to more than 20,000 digital twin scenarios of 15-20 seconds in length and up to 10 Hz sample rate.

To get the conditions (*e.g*, depth, segmentation) in the simulation domain (*Sim Conditions*), we replay the ego vehicle trajectory recorded in the scene record in the simulator, and capture the simulated sensor data from the front view camera. To achieve pixel-wise alignment between the digital twins, we calibrated the simulator sensors according to the camera parameters (intrinsics and extrinsics) recorded in the nuPlan dataset for each scenario. Then we capture the Sim Conditions synthesized using Panda3D engine based on OpenGL rendering backend, which includes depth map, semantic map, instance map, rendered RGB, and top-down view. The Sim-LWA is constructed from these Sim Conditions.

**Annotation**. We annotate our digital twin scenarios to obtain realistic conditions (*Real Conditions*) using pre-trained foundation models. Given a realistic front camera view from the nuPlan sensor data, we use DepthAnything2 (Yang et al., 2024) to obtain the depth map, SegFormer (Xie et al., 2021) for the segmentation map, GroundingDino (Liu et al., 2023c) + SAM2 (Ravi et al., 2024) for instance map, and Llama 3 (Grattafiori et al., 2024b) for text descriptions. All of the Real Conditions align with the Sim Conditions in the pixel space, which enables us to construct pairwise training data for Sim2Real transfer described below.

## B.2   D3SIM TRAINING

Given the digital driving scenario with Sim and Real conditions, we construct our training dataset for the second and third stages. We first split the conditions into three world layers according to the segmentation mapping in Table 5, the visibility mask $\mathcal{V}$ for each layer is derived as the pixels that belong to the corresponding semantic classes. Then we construct the concatenated conditions along the channel dimension, and construct the Sim-LWA and Real-LWA according to the predefined preserved and editable layers. The whole construction process is shown in Figure 9, where we only include the segmentation map as a condition for demonstration.

| World Layer | Semantic Classes | | | | | |
|---|---|---|---|---|---|---|
| Traffic Participants $\mathcal{L}^d$ | CAR | TRUCK | BUS | PEDESTRIAN | BICYCLE | MOTORCYCLE |
| Map Layout $\mathcal{L}^l$ | ROAD | CROSSWALK | SIDEWALK | FENCE | TRAFFIC_LIGHT | TRAFFIC_SIGN |
| Background Layer $\mathcal{L}^b$ | SKY | TERRAIN | BUILDING | VEGETATION | WALL | POLE |

Table 5: **Semantic mapping for world layers**.

**Data Diversity**. To demonstrate the layout diversity of our D3Sim dataset, we visualize the top-down view of various driving scenarios in Figure 10, including unprotected cross-turn, dense vehicle interactions, pickup/dropoff area, and following vehicle.

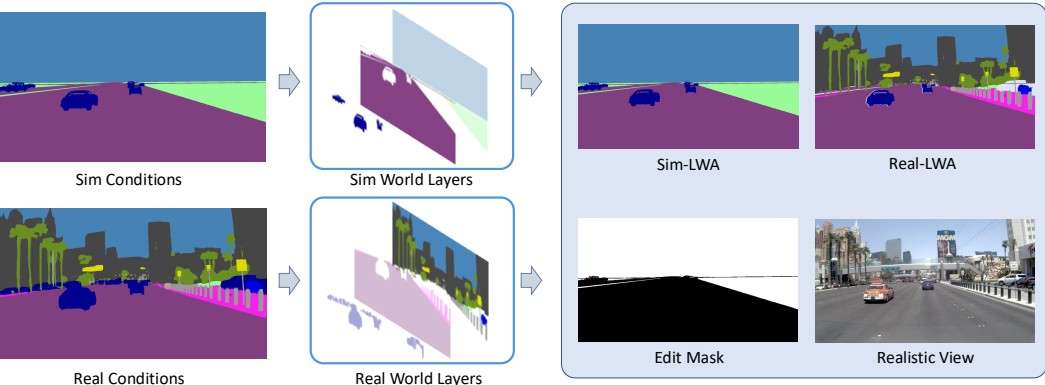

Figure 9: **Layered World Abstraction construction pipeline**.

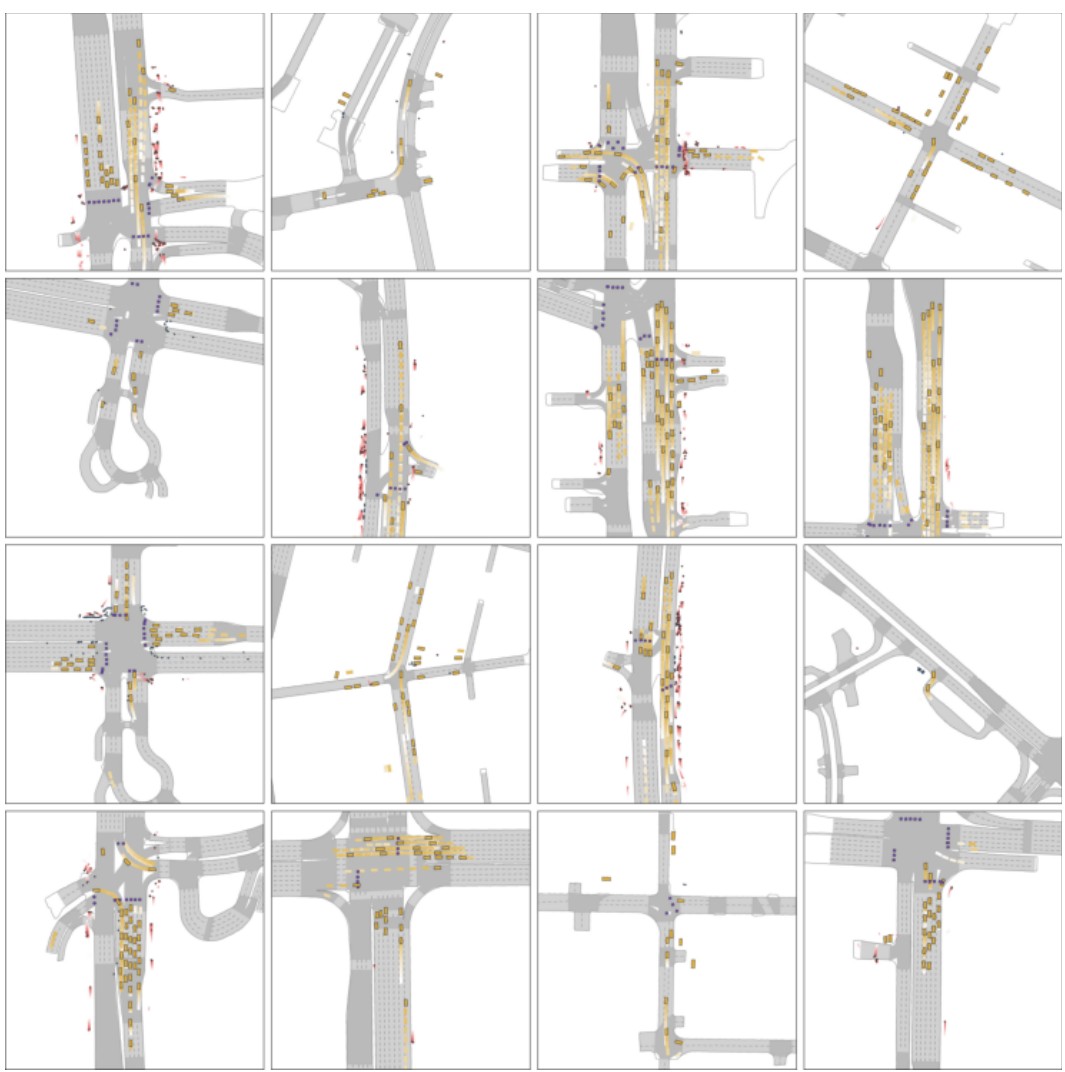

Figure 10: **Diverse Scene Layouts from D3Sim**. Ego vehicle and other traffic participants are marked with colored rectangles.

## B.3 D3SIM VALIDATION

We derive the D3Sim validation dataset from digital twins to evaluate controllability and visual quality. Despite driving scenarios from the nuPlan dataset reflecting real-world distribution, its

complex object layout and interaction result in overlapping and occlusion, which hinders the accurate evaluation of controllability. Therefore, we construct an automated derivation pipeline to obtain scene records with diverse static layouts and unambiguous traffic participants' placement.

Given a scene record, we first filter out occluded and non-visible objects according to the ego agent's view. Then we derived different variations of the scene by sampling from the combination of the objects. This process ensures that all objects in the sampled scenarios are clearly visible for controllability evaluation. We also randomly generate multiple diverse prompts for each sample, which are used as the text condition for the world model to evaluate its visual quality and diversity.

### B.4 D3SIM VIDEO

To extend our pipeline to video-based world models, we construct the video dataset with paired Sim and Real video conditions, along with realistic perspective view videos. The dataset contains around 20,000 video clips, each corresponding to a 15-20 second driving scenario captured at a 10 Hz sample rate. We follow the process described in Sec. B.2 to construct the Sim-LWA, Real-LWA, and the edit mask for each frame, then concatenate the processed frames into videos. These paired videos are used to train the second stage of the Dreamland-Video pipeline.

## C IMPLEMENTATION DETAILS

### C.1 DREAMLAND IMPLEMENTATION DETAILS

We follow ACE++ (Mao et al., 2025) to train our Stage-2 instructional editing model. Inspired by REVA (Kara et al., 2024), we vertically concatenate the depth and segmentation maps, each of size $512 \times 288$, into a single image of shape $512 \times 576$. We find that the instructional editing model can accurately generate structures in both the upper and lower conditional maps simultaneously, without any additional loss. We compare this approach with SimGen's method, which encodes the segmentation and depth maps into the Red and Blue channels, respectively. Empirically, we find that our approach performs better, as it is more aligned with the generative model's pre-training setting.

For the Stage-3 conditional generation model, we initialize it from the FLux Depth (Black Forest Labs, 2024) model and only train the linear projection layer $\xi$ to incorporate additional segmentation maps as conditioning inputs. To achieve this, we increase the input channels of the projection layer from 128 to 192, while keeping the output channel count unchanged. We also experimented with using LoRA to finetune the Stage-3 model; however, we empirically found that this approach tends to overfit to the finetuning dataset and degrade the model's world knowledge. Therefore, we only finetune the linear projection layer to preserve world knowledge while maintaining controllability.

### C.2 DREAMLAND-VIDEO IMPLEMENTATION DETAILS

In Stage-2, we fine-tune the Cosmos-Predict1-7B (Agarwal et al., 2025) Text-to-Video diffusion model into an instructional video-editing model. The training uses our D3Sim video dataset for 10K iterations with a batch size of 8. Following standard practice in image editing models, we initialize from the pretrained generative model and concatenate the extra conditioning latents with the noise latent along the channel dimension.

For Stage-3, we employ a multi-condition video diffusion model, Cosmos-Transfer1-7B (Agarwal et al., 2025), without finetuning. Leveraging this state-of-the-art model, Dreamland-Video can produce videos from 1K up to 4K resolution and as long as 121 frames. It is worth noting that, by swapping in other pretrained world models, the Dreamland-Video framework can be extended to even longer or fully autoregressive video re-rendering. We leave further exploration of this promising direction to future research.

## D EXPERIMENTS

### D.1 EVALUATION METRICS

We evaluate Dreamland from two key aspects: Generation Quality and Controllability.

| Methods | Stage3 Input | Stage3 Model | Image Quality | Controllability | |
|---|---|---|---|---|---|
| | | | FID ↓ | si-RMSE ↓ | mIoU ↑ |
| Dreamland (Variations) | Sim-LWA | Frozen SDXL | 106.80 | 0.806 | 0.505 |
| | Sim-LWA | Frozen SD3 | 78.30 | 0.808 | 0.638 |
| | Sim-LWA | Frozen Flux | 76.21 | 0.737 | 0.623 |
| | Sim-LWA | Finetuned Flux | 106.93 | 0.641 | 0.595 |
| | Real-LWA | Frozen SDXL | 78.99 | 0.744 | 0.699 |
| | Real-LWA | Frozen SD3 | 51.65 | 0.673 | 0.747 |
| | Real-LWA | Frozen Flux | 45.19 | 0.640 | 0.786 |
| Dreamland | Real-LWA | Finetuned Flux | 44.61 | 0.647 | 0.791 |

Table 6: **Ablation: Real-LWA** improves generation fidelity to real driving scene.

**Generation Quality**. For generation quality, we use Fréchet Inception Distance (FID) (Heusel et al., 2017). It measures the distribution distance of features between generated and original frames in the dataset. The features are extracted using a pre-trained Inception model. For quantitative results and comparisons, FID is evaluated on 10,000 samples from DIVA-Real (Zhou et al., 2024b) to reflect the real-world driving distribution if not explicitly specified.

**Controllability**. For the quantitative evaluation of adherence to control input conditions, we transform the generated samples into shared representation spaces by applying depth estimation and semantic segmentation operations. We then compare the transformed representation with the original conditions captured from the simulator in the preserved region to measure the alignment between the generated world and the simulation.

For depth alignment, we compute the scale-invariant Root Mean Squared Error (si-RMSE) (Eigen et al., 2014) between the depth map from the simulation and transformed by DepthAnythingV2 (Yang et al., 2024), where a lower value represents better alignment. For semantic alignment, we compute the mean Intersection over Union (mIoU) between the segmentation map from the simulation and the generated frames derived using SegFormer (Xie et al., 2021). Higher value means better alignment.

## D.2 MORE ABLATION

**Scalability**. Dreamland's simple pipeline design unlocks the potential of better image generation capability and controllability by upgrading the Stage-3 model to stronger image generative foundation models. Supported by Table 1, Table 8, and user study results in Table 7, Dreamland could constantly combine with state-of-the-art conditional generation models to enhance the existing pipeline.

**Real-LWA**. We ablate the effectiveness of our Stage-2 model design by forwarding Stage-3 with either Sim-LWA or Real-LWA . As illustrated in Table 6 and Figure 11, using Real-LWA consistently improves the FID, indicating that the generated content achieves better fidelity to real driving scenes. We also observe similar results in experiments on the nuScenes dataset, as recorded in Table 8. These observations highlight the necessity of using Real-LWA in a hybrid pipeline for scene creation.

## D.3 QUALITATIVE RESULTS

**Alignment to text prompts**. Dreamland is capable of rendering the simulator into realistic frames while taking user-provided text prompts. As shown in Figure 17, Dreamland can synthesize realistic scenes with different locations, weather, times, or styles, demonstrating world knowledge preservation.

**Alignment to simulator conditions**. We show more qualitative results of the Dreamland in Figure 18. Dreamland can re-render complex driving scenes into realistic frames with preserved scene layout.

**Flexible simulator control with LWA**. Dreamland's LWA design can dynamically divide the simulator world into preserved and editable regions based on user instructions. This allows users to control specific vehicles or road layouts by adding them to or removing them from the preservation

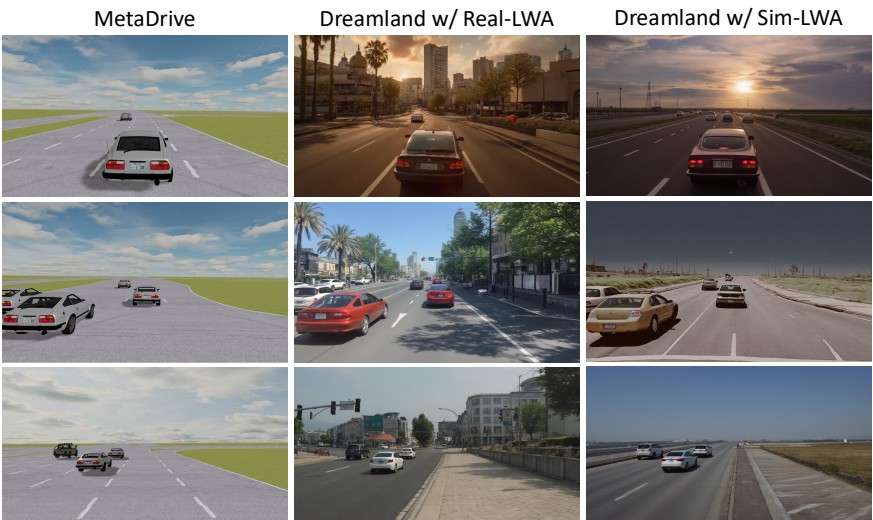

Figure 11: **Ablation: Dreamland with Real-LWA and Sim-LWA**. Dreamland using Sim-LWA as the Stage-3 input generates driving scenes with depth identical to the MetaDrive conditions, but loses fidelity to real driving scenes.

region. In Figure 13, we show that Dreamland's output closely aligns with the user-defined Sim-LWA and effectively respects the user-specified editable regions, indicating that fine-grained simulator control can be achieved through this hybrid approach.

**Extending to other simulators**. Dreamland's LWA design is flexible to extend to other types of simulators. We applied Dreamland on MetaUrban (Wu et al., 2025b) and Isaac Sim (NVIDIA). For MetaUrban, we follow the same setting of the LWA semantic mapping and directly integrate with Dreamland trained on D3Sim. As shown in Fig 12 (a), Dreamland showcased strong zero-shot performance on other types of mobility simulators.

To integrate Dreamland with Isaac Sim, we re-train the Stage-2 model with additional data and use the Frozen Flux-depth conditional model as the Stage-3 model. We use Qwen-Image (Wu et al., 2025a) to generate 3K images with target objects for the robotic manipulation task (*e.g* toys and cups on a table), and we edit the generated images with Qwen-Image-Edit (Wu et al., 2025a) for object addition or background variation. We then construct the LWA for the Stage-2 model following the pipeline in D3Sim training, with semantic mapping designed for manipulation tasks. After the training, our Stage-2 model could vary the structure of a manipulation scene by changing the background or adding additional objects, as demonstrated in Figure 12 (b). Such synthetic data could serve as diverse digital cousins to bootstrap the policy training for robot manipulation tasks.

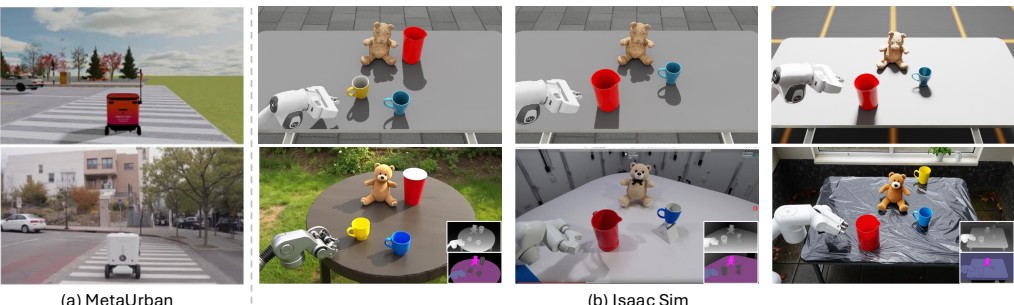

(a) MetaUrban      (b) Isaac Sim

Figure 12: **Applying Dreamland on MetaUrban and Isaac Sim.** Dreamland can re-render urban mobility scenes into realistic frames with preserved scene layout, and generate realistic digital cousin for manipulation tasks by changing background(*b left, and b middle*) or adding other objects (*b right*).

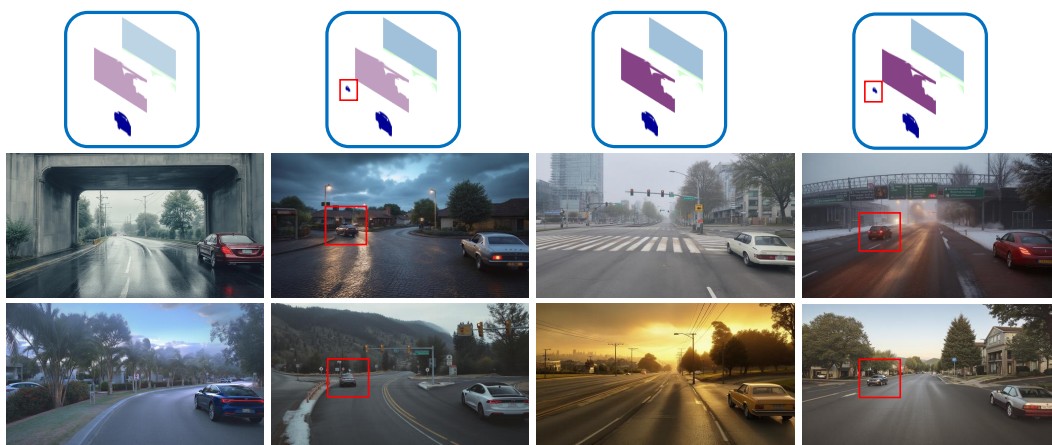

Figure 13: **Flexible control with LWA**. *Top*: Visualization of Sim-LWA, where layers/objects not under user control are made transparent. *Bottom*: Dreamland output. Users can choose to control specific vehicles or the road layout by adding them to or removing them from the preservation region. We use a red bounding box to highlight the small car that the user prefers to control in both the LWA and Dreamland outputs.

| Baseline | Image Quality | | | | Controllability | | | |
|---|---|---|---|---|---|---|---|---|
| | Ours Better | Opponent Better | Both Good | Both Bad | Ours Better | Opponent Better | Both Good | Both Bad |
| Dreamland w/ Sim-LWA | **36.6** | 32.0 | 17.1 | 14.3 | **52.6** | 18.9 | 24.6 | 4.0 |
| Panacea | **88.5** | 3.3 | 2.7 | 0.5 | **79.1** | 7.7 | 4.4 | 8.8 |
| MagicDrive | **90.9** | 2.3 | 4.6 | 2.3 | **63.4** | 14.3 | 8.0 | 14.3 |
| SimGen | **95.4** | 0.6 | 0.6 | 3.4 | **71.4** | 10.3 | 6.3 | 12.0 |
| Dreamland (*Frozen SDXL*) | **90.9** | 0.6 | 3.4 | 5.1 | **73.7** | 9.1 | 10.3 | 6.9 |
| Dreamland (*Frozen SD3*) | **83.4** | 4.6 | 6.3 | 5.7 | **86.9** | 1.7 | 1.1 | 10.3 |
| Dreamland (*Frozen Flux*) | 33.7 | **44.0** | 14.3 | 8.0 | **46.3** | 31.4 | 12.6 | 9.7 |

Table 7: **User Study**. We study the user preference by comparing Dreamland and its baseline methods. The result suggests that Dreamland achieved strong visual generation quality and outperforms all other baselines in alignment with simulator conditions.

**User study**. To fully evaluate our method, we conducted a survey to study user preference between Dreamland and its baseline methods and variations. We randomly selected 50 samples from our D3Sim validation dataset and compared the generated figures between ours and a baseline method regarding image quality and simulator controllability. Users are asked to choose from four options, including: "Image A is better", "Image B is better", "Both are equally good", and "Both are equally bad". We recruited 25 users, each of whom evaluated 7 different samples, resulting in a total of 175 user votes for each comparison between our method and the baseline. Figure 14 shows our survey template and Table 7 reports the user preference between our methods and selected baselines.

In the comparison between Dreamland and Dreamland use Sim-LWA as the input for Stage-3, noted as Dreamland w/ Sim-LWA, using Real-LWA leads to 4.6 and 33.7 absolute percentage in image quality and controllability, showcasing the importance of our Stage-2 model. It is also worth noting that Dreamland largely outperforms the previous state-of-the-art method SimGen (Zhou et al., 2024b), achieving $95.4\%$ user preference in image quality and $71.4\%$ in simulator controllability.

## D.4 COMPARISON WITH OTHER BASELINES

**Experimental Setup.** We compare Dreamland with previous generative models for driving scene generation, including BEVGen (Swerdlow et al., 2024a), BEVControl (Yang et al., 2023), MagicControl (Gao et al., 2023), Panacea (Wen et al., 2024), DrivingDiffusion (Li et al., 2023b), and SimGen (Zhou et al., 2024b). SimGen shares the same setup as ours, employing a simulator to first render scene records and then using a generative model to re-render the simulator frames. The other baselines directly encode driving scene maps and generate driving scenes based on them. We conduct

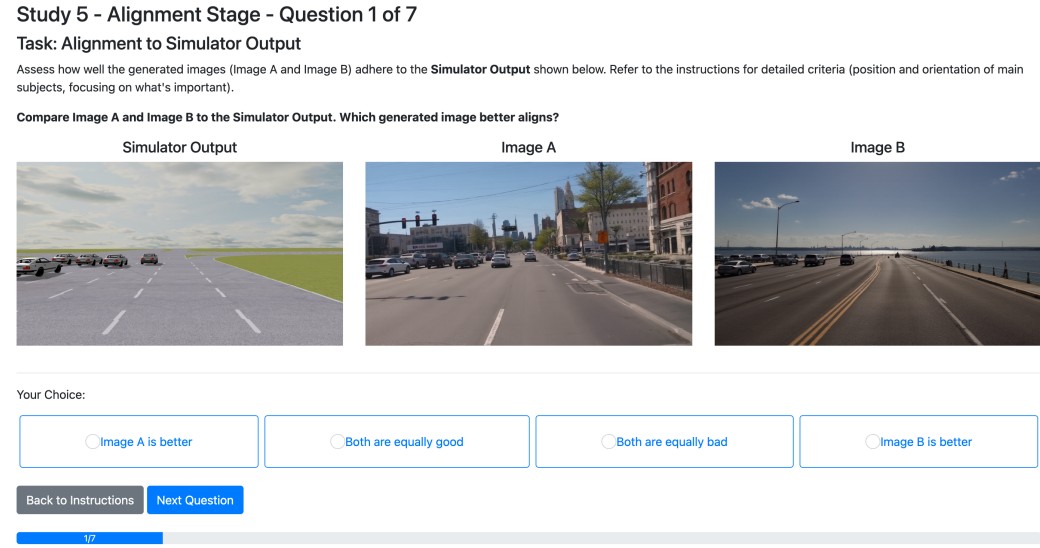

Figure 14: **Screenshot of user study templates.** Users are asked to select between "Image A is better", "Image B is better", "Both are equally good", and "Both are equally bad" options. We randomly shuffle the order of images in the user study.

the experiment using the 6K spatial conditions from the nuScenes validation dataset and compute the FID with source images from nuScenes or DIVA-Real.

**nuScenes FID Assessment of Generative Models.** In recent years, it has been common practice for generative models in autonomous driving tasks to report FID scores based on natural driving images, such as nuScenes dataset (Caesar et al., 2019). However, recent large-scale foundation models (Podell et al., 2024; Esser et al., 2024; Black Forest Labs, 2024) are often pre-trained on datasets rich in high-quality, visually aesthetic images. This leads them to generate stylized, artistic outputs that differ from naturally captured driving scenes. As a result, evaluating generative models for autonomous driving using nuScenes FID has become more and more questionable. Zhou et al. (Zhou et al., 2024b) observed that generative models trained on the DIVA-Real dataset demonstrate noticeably improved visual fidelity. Therefore, we compute FID using DIVA-Real, since it captures high-quality, aesthetically realistic driving scenes aligned with modern generative models.

While previous state-of-the-art models achieve low FID scores on nuScenes, their FID scores on DIVA-Real are significantly higher. For example, both Panacea (Wen et al., 2024) and SimGen (Zhou et al., 2024b) achieve nuScenes FID scores around 16, but their FID scores on DIVA-Real exceed 60. We validate the effectiveness of using FID on DIVA-Real through a human evaluation study, with results reported in Table 7. In our user study, 95.4% of participants agreed that Dreamland produces higher visual quality compared to SimGen, and over 80% preferred Dreamland over the Dreamland variants using SDXL and SD3. This aligns with the FID scores on DIVA-Real, where Dreamland significantly outperforms the aforementioned methods. It is also worth noting that when Dreamland and Dreamland (*Frozen Flux*) achieve similar FID scores (44.61 vs. 45.19), human preference is also consistent with the scores, with winning rates of 33.7% and 44.0%, respectively.

**Quantitative results**. Table 8 presents a comparison between Dreamland and previous state-of-the-art models for autonomous driving tasks. Under the actual inference setting, Dreamland achieves the lowest FID on DIVA-Real (44.61), outperforming the previous best method, MagicDrive, by 12.8%, demonstrating the strong capability of Dreamland in generating realistic driving scenes. Notably, Dreamland using ground truth condition maps (e.g., depth and segmentation maps) achieves a similar FID score to Dreamland with Real-LWA , highlighting the effectiveness of our LWA design.

| Method | Stage3 Input | Stage3 Model | Image Quality | |
| --- | --- | --- | --- | --- |
| | | | FID - nuScenes | FID - DIVA Real |
| BEVGen (Swerdlow et al., 2024b) | | | 25.50 | - |
| BEVControl (Yang et al., 2023) | | | 24.90 | - |
| MagicDrive (Gao et al., 2023) | | | 16.60 | 51.20* |
| Panacea (Wen et al., 2024) | | | 16.96 | 61.83* |
| DrivingDiffusion (Li et al., 2023b) | | | 15.90 | - |
| SimGen (Zhou et al., 2024b) | | | 15.60 | 68.20* |
| | GT Conditions | Frozen SDXL | 54.20 | 65.47 |
| | GT Conditions | Frozen SD3 | 43.68 | 50.01 |
| | GT Conditions | Frozen Flux | 43.22 | 44.63 |
| | GT Conditions | Finetuned Flux | 42.36 | 42.27 |
| Dreamland (Variations) | Sim-LWA | Frozen SDXL | 93.66 | 100.57 |
| | Sim-LWA | Frozen SD3 | 55.49 | 59.92 |
| | Sim-LWA | Frozen Flux | 52.66 | 51.68 |
| | Sim-LWA | Finetuned Flux | 51.35 | 52.47 |
| | Real-LWA | Frozen SDXL | 68.69 | 78.99 |
| | Real-LWA | Frozen SD3 | 48.73 | 51.65 |
| | Real-LWA | Frozen Flux | 46.00 | 45.19 |
| Dreamland | Real-LWA | Finetuned Flux | 47.93 | 44.61 |

Table 8: **Quantitative comparison with baseline methods**. All methods are evaluated using the spatial conditions from the nuScenes validation dataset, and the FID is computed with source images from nuScenes or DIVA-Real. For existing baselines, we use the reported FID on the nuScenes dataset from their respective publications, and compute the FID on DIVA-Real when open-source checkpoints are available. We mark scores reproduced by us with an asterisk (*).

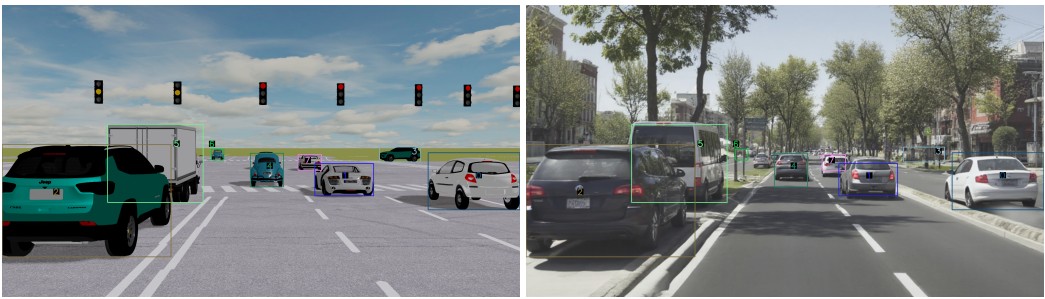

(a) MetaDrive-rendered Observation.      (b) Dreamland-rendered Observation.

Figure 15: **Example questions**. `Suppose our current speed is slow(0-10 mph), and we perform action "KEEP_STRAIGHT" for 1.0 seconds. Will we run into object <2>, provided that it remains still? Select the best option from: (A) Yes; (B) No.` The answer is (B)

## D.5 EMBODIED AGENT TRAINING

We explain the experimental setting in greater detail in this section. To curate the three training sets for this task mentioned before, we use a shared set of traffic scenarios leveraging the Scenarionet (Li et al., 2023a) data platform. These scenarios are rendered with both the MetaDrive (Li et al., 2022) simulator and Dreamland(see Figure 15 for illustration). For each scenario, we generate questions using the MetaVQA (Wang et al., 2025) annotation tools, and randomly sample 1000 VQA tuples for each observation domain. We use the InternVL2 (Chen et al., 2024) codebase for the LoRA fine-tuning of pre-trained InternVL2-8 B. We train the model for two complete epochs and evaluate the model's performance on a test set, which is generated using nuScenes (Caesar et al., 2019) and Waymo (Sun et al., 2020) datasets. The test set also comprises real images(from nuScenes) and simulator-rendered images(from Waymo) for holistic evaluation of the learned model under different visual domains. The answer parsing and metrics calculation directly use the toolkit provided by the MetaVQA codebase. Refer to the main paper for our results.

| Method | Visual Quality | | | | Controllability | |
|---|---|---|---|---|---|---|
| | f-30 FID ↓ | f-60 FID ↓ | f-90 FID ↓ | f-120 FID ↓ | si-RMSE ↓ | mIoU ↑ |
| Dreamland-Video | 90.74 | 88.93 | 91.51 | 88.78 | 0.659 | 0.717 |

Table 9: **Quantitative Results of Dreamland-Video.** Our pipeline preserved strong video generation and outstanding condition-following capabilities while adapting to our world representations.

### D.6    DREAMLAND-VIDEO RESULTS

**Quantitative Results.** We evaluate the Dreamland-Video generation capability with the FID on D3Sim dataset using all extracted frames at a specific timestep from 800 generated videos. As shown in Table 9, our approach delivers consistent generation results across different frames.

**Qualitative Results.** We provide additional video results in the supplementary materials under the file name *dreamland_video_supp.mp4*, which includes four sections: (1) Simulator-controlled scene generation, (2) Diverse text-controlled scene generation with the simulator (3) Multi-view simulator-controlled scene generation. (4) Simulator-controlled safety-critical driving scene generation, and (5) Diverse safety-critical driving scene generation with the simulator. While the video was originally generated at high resolution (4K), it has been compressed to meet the 100MB file size limit, resulting in lower visual quality than the original output. Nevertheless, Dreamland-Video demonstrates strong performance in world creation with a physical simulator and generative models. Its zero-shot capability to generate safety-critical scenarios allows efficient sampling of dangerous driving behaviors, laying the foundation for training autonomous agents with safe behavioral responses.

**Extending to Multi-View Video Generation**. As shown in Figure 16, Dreamland supports multi-view generation by integrating a multi-view model, Cosmos-Drive-Dreams (Ren et al., 2025), as part of the Stage-3 model. Different from (Ren et al., 2025), Dreamland could support fine-grained scene customization by editing the scenes in the specific view without changing the global structure.

## E    LIMITATION

Dreamland introduced an additional editing model to the current hybrid pipeline, which led to a longer inference time. Also, the high-quality simulator and real-world paired data are expensive to annotate, which may limit the performance of the instructional editing model.

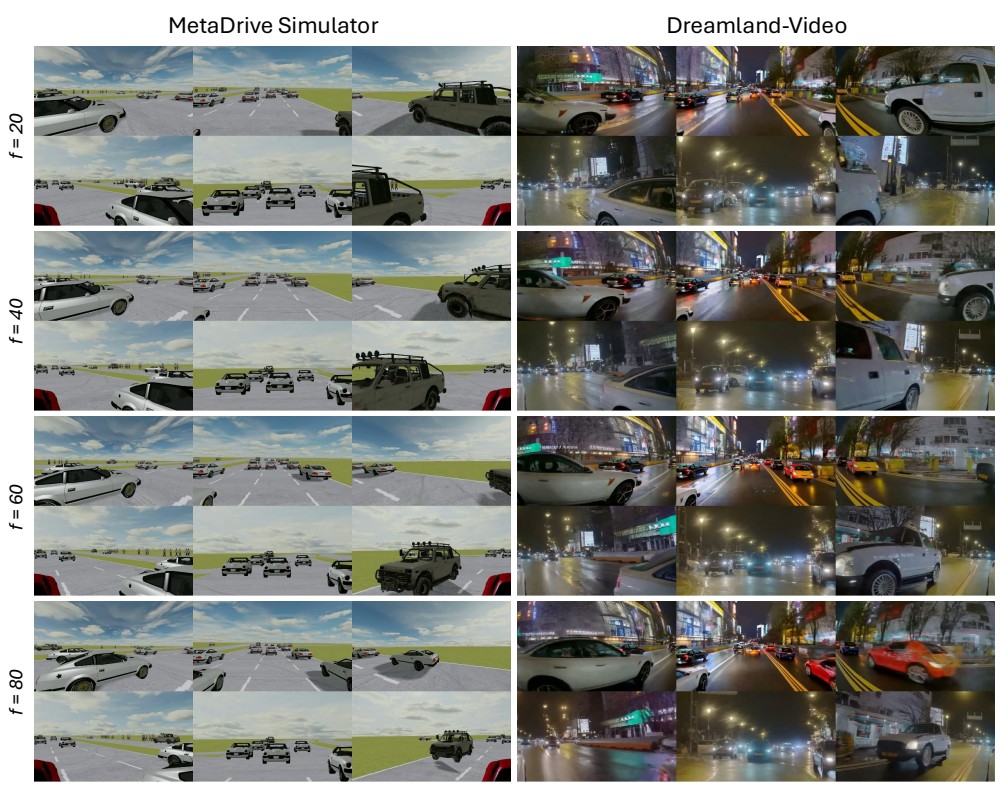

Figure 16: **Dreamland with Multi-View Video Generation**. By integrating with a multi-view video diffusion model as Stage-3 model, Dreamland could extend its controllability and its flexibility to multi-view generation.

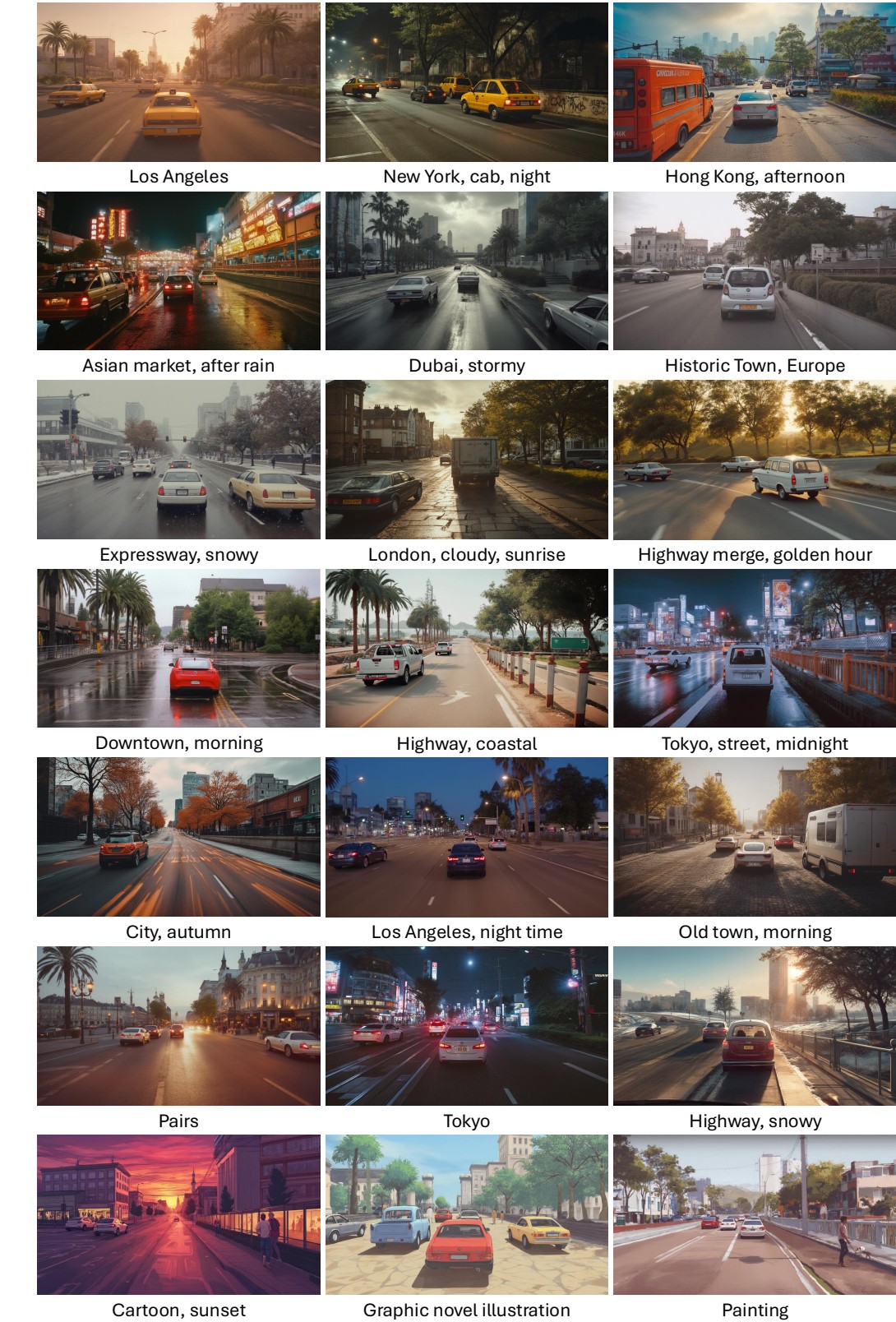

Figure 17: **Text-grounded scene creation with Dreamland**. Dreamland could create diverse and visually realistic driving scenes by preserving the world knowledge in the pre-trained generative models.

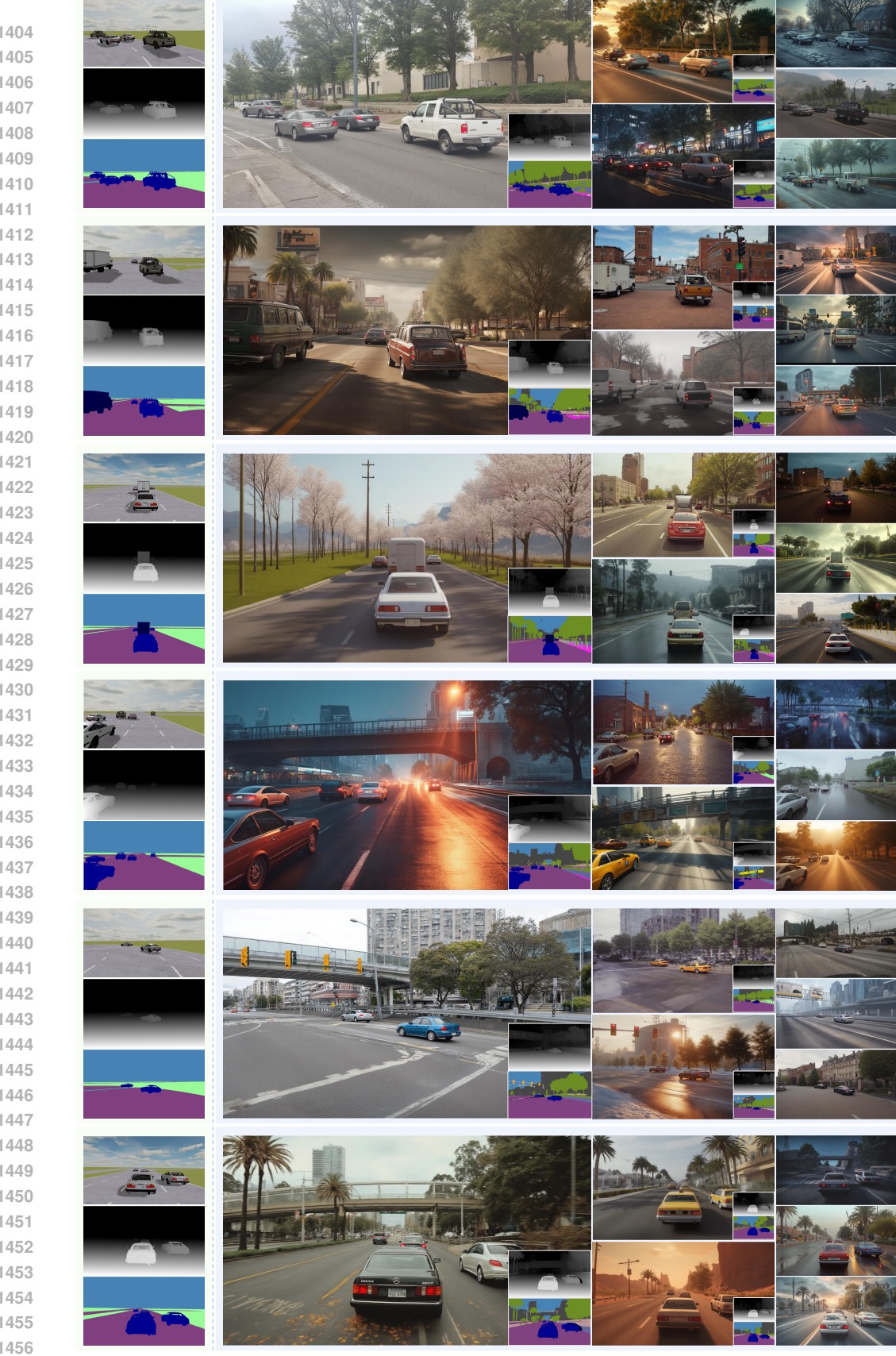

Figure 18: **Rendering diverse appearances aligned to the simulator's conditions.** Dreamland can re-render complex driving scenes into realistic frames with preserved scene layout.

