# OpenReview forum: "Dreamland: Hybrid World Creation with Simulator and Generative Models"
_ICLR.cc/2026/Conference — Submitted to ICLR 2026_

### Official Review · Reviewer_FF93 · 2025-10-25

**Soundness:** 3
**Presentation:** 3
**Contribution:** 3
**Rating:** 6
**Confidence:** 4

**Summary:**

The paper proposes Dreamland, a hybrid world generation framework that integrates physics-based simulators with large-scale generative models for controllable world generation especially for driving scenarios. The paper proposes Layered World Abstraction (LWA), which is a core contribution bridging the simulator domain and real-world domain to achieve both physical controllability and photorealistic fidelity.

Aditionally, the paper proposes D3Sim, a curated large-scale dataset containing paird simulated and real-world driving scenarios for training and evaluation. Experiments on image quality, controllability, and downstream tasks demonstrate that Dreamland achieves significant improvements over previous baselines such as SimGen, achieving 52.3% lower FID and 17.9% better controllability. Dreamland also generalizes to new simulators and video generation with minimal retraining.

**Strengths:**

1. Strong quantitative gains: significant improvements in FID and controllability, outperforming state-of-the-art models.
2. Novel hybrid formulation: clear, elegant bridge between simulator and generative domains via LWA.
3. Scalability: the pipeline can pulg in newer pretrained models with minimal cost.

**Weaknesses:**

1. Compute and efficiency not detailed: training and inference costs for different model stages are not fully reported.
2. The paper could analyze the stability or convergence of LWA-Sim2Real adaptation more deeply.
3. While the framework is claimed to be general, experiments are mostly in driving scenes. A short cross-domain demo would strengthen the claim of universality.

**Questions:**

1. How is temporal consistency handled in Dreamland beyond perfram LWA conditioning?
2. Is there any failure cases when the pretrained generative model's world knowledge conflicts with simulator conditions?

---

> ### Author Response · Authors · 2025-11-21
>
> We thank the reviewer for the constructive feedback. We are encouraged that the reviewer found our Layered World Abstraction (LWA) “novel, clear, and elegant”, and our method achieves “strong gains” and is scalable. We address the concerns as follows.
>
> ### Weakness
>
> > Q1: Training and inference costs
>
> We thank the reviewer for pointing this out and report the training and inference costs for each model stage below, measured on an A6000 GPU. It is worth noting that the inference time of each stage is determined by the design and size of the pre-trained model, and Dreamland does not introduce any additional inference cost at any stage.
>
> | Model Stage             | Inference Time | Training Time  |
> | ----------------------- | -------------- | -------------- |
> | Dreamland Stage-2       | 8s             | ~96 GPU hours  |
> | Dreamland Stage-3       | 6s             | ~32 GPU hours  |
> | Dreamland-Video Stage-2 | 230s           | ~960 GPU hours |
> | Dreamland-Video Stage-3 | 1170s          | \-             |
>
> > Q2: Stability/convergence of Stage-2
>
> Thanks for the great suggestion! We additionally analyze the convergence of Stage-2 models with limited paired data for the same number of steps and evaluate the pipeline to obtain the results shown below. While training on limited paired data compromises the controllability to a certain extent, the visual quality remains competitive thanks to our flexible design.
>
> | Setting          | FID   | si-RMSE | mIoU  |
> | ---------------- | ----- | ------- | ----- |
> | 25% Paired Data  | 49.75 | 0.646   | 0.747 |
> | 50% Paired Data  | 50.93 | 0.629   | 0.74  |
> | Full Paired Data | 50.58 | 0.646   | 0.791 |
>
> > Q3: Cross-domain demo
>
> Yes, we agree that cross-domain applicability is important. In Sec. 5.4 and App. D.3, we already demonstrated that Dreamland can be applied to indoor scenes beyond driving. While Dreamland has strong potential across other domains, we will further clarify this point and adopt a more measured tone on our claim regarding universal hybrid world creation in the camera-ready version.
>
> ### Questions
>
> > Q1: Temporal Consistency
>
> As described in L956–L966, we finetune the text-to-video model into a video-to-video editing model using LoRA. The resulting temporal consistency comes from the strong temporal priors already learned by the T2V model.
>
> > Q2: Failure Cases
>
> Stage-1 simulator outputs are physically grounded and therefore rarely conflict with the Stage-3 pretrained generative model’s prior knowledge. In practice, Stage-3 handles corner cases well when provided with the condition maps, as demonstrated in the safety-critical scene in the supplementary video.

---

> ### Comment · Reviewer_FF93 · 2025-11-25
>
> Thank you! My questions and concerns are fully addressed. Great work! I will raise my score to 8. Looking forward to seeing your cross-domain demo in the future!

---

> > ### Author Response · Authors · 2025-11-28
> >
> > Thank you for your suggestion and for raising the score for acceptance!

---

### Official Review · Reviewer_m9s3 · 2025-10-28

**Soundness:** 3
**Presentation:** 3
**Contribution:** 2
**Rating:** 4
**Confidence:** 3

**Summary:**

This paper introduces a pipeline to bridge simulators to generative models by designing layered world abstraction, which can be used to augment the training of embodied agent. The full pipeline utilizes editing models and conditional generative models for generation. This paper utilizes this pipeline to construct the D3Sim dataset for training and benchmarking Sim2Real transfer between simulators and generative models.

**Strengths:**

- This paper proposes layered world abstraction to bridge the gap of semantic maps generated by simulators and the ones generated by generative models.

- This paper constructs D3Sim dataset to facilitate Sim2Real transfer.

- This paper conducts evaluation on downstream tasks to assess effectiveness of DreamLand.

**Weaknesses:**

- The proposed pipeline heavily relys on models introduced in previous works with minor modifications. We expect authors to clarify more details about the innovation of this paper.

- This paper only conduct the quantative comparsion with other baseline methods based on image quality, except for SimGen, which is incomplete. We expect more results on controllability of other baseline methods, such as Panacea and MagicDrive.

**Questions:**

see above.

---

> ### Author Response · Authors · 2025-11-21
>
> We thank the reviewer for the constructive feedback. We are encouraged that the reviewer found our Layered World Abstraction (LWA) “bridges the gap between simulator and generative models”. We are glad that the reviewer recognized our D3Sim dataset’s contribution and our downstream evaluations. We address the concerns as follows.
>
> ### Weakness
>
> > Q1. Clarification on innovation
>
> The proposed pipeline heavily relies on models introduced in previous works with minor modifications. We expect authors to clarify more details about the innovation of this paper. Indeed, our main contribution is introducing LWA as a flexible yet controllable bridge between simulators and generative models. While SimGen and LucidSim explore this direction, LucidSim relies on fully rendered scenes without hallucination, and SimGen sacrifices fine-grained control over simulator content. Thus, effectively bridging simulators and generative models for high-quality, controllable digital-twin/cousin data **remains an important but unsolved challenge**.
>
> We find that keeping Stage-1 condition maps precise and selectively edited is key to preserving grounded structure, which directly motivates introducing LWA to achieve this. As shown in Figure 8, LWA avoids hallucinating extra vehicles (left) and preserves correct pedestrian shapes (right). This controllability supports diverse applications, and Section 5.4 shows that Dreamland-generated data improves downstream detection and VLM performance.
>
> For Stage-2 and Stage-3, we follow common settings in the Generative AI community, highlight the generalizability of Dreamland and enable integration and simple adaptation to future, more potent pre-trained instructional editing and conditional generation models.
>
> > Q2: More controllability evaluation
>
> We thank the reviewer for mentioning this and further evaluating the controllability of baseline methods on the nuScenes dataset and report the results below. Our Dreamland achieves consistent improvements across depth and semantic alignment.
>
> | Method     | si-RMSE | mIoU  |
> | ---------- | ------- | ----- |
> | SimGen     | 0.837   | 0.480 |
> | Panacea    | 0.695   | 0.489 |
> | MagicDrive | 0.679   | 0.553 |
> | Dreamland  | 0.668   | 0.571 |

---

### Official Review · Reviewer_5jdQ · 2025-10-30

**Soundness:** 2
**Presentation:** 3
**Contribution:** 3
**Rating:** 4
**Confidence:** 4

**Summary:**

This paper introduces Dreamland, a simulation-aware driving scene generation model, which at its core of design is a layered world abstraction representation that bridges grounded simuations and the real world.

At first, it constructs a grounded scene in the simulator and derive its layered form of abstraction, which is in the visual perspective. Then, this layered representation, containing a traffic-participent layer, a map-layout layer, and a background layer, is then processed by a trained instructional editing model, which hallucinates details in the background layer - making it closer to real-world semantics. At last, this refined layered representation is serving as condition to a fine-tuned photorealistic generation model.

They also include a dataset for training and evaluation, which is derived from another public driving scene dataset. Experiments and further extensions to other simulators and taks are also discussed and included.

**Strengths:**

**Interesting abstraction design.**
Although the layered scene concept can be originated to another paper Kimera [1], this layered world abstraction (LWA) in the autonomous driving literature can be seen as new. How this paper proposes to use this LWA as conditional signal for controlability is interesting.

**Clear paper writing and structuring.**
This paper is written in an easy-to-understand manner and the procedural pipeline is clear with mathematical expressions. Details on how each model is trained and how data is curated is also clear for reproducibility.

**Wide applicability for other domains.**
The core method and main experiments / applications remain autonomous driving centric, but the authors also provided a proof-of-concept verification on other domains like robotic manipulation in sec. 5.2 and supp. D.3. This shows the LWA representation is suitbale not solely in outdoor scenes but also indoor scenes, and the object-layout-background seems to be somehow transferable.

```
[1] Kimera: from SLAM to Spatial Perception with 3D Dynamic Scene Graphs.
```

**Weaknesses:**

**Limited real-world verification.**
In sec. 5.4 and supp. D.5, the authors have fine-tuned a VLM for real-image-VQA, trying to show improved learning for enriched perception. But unfortunately, IMHO, this has nothing to do with *embodied agent*. VQA tasks are neither related to embodiments nor agents, which makes this expression inappropriate. The relevance to embodied-agent training remains speculative.

**Limited conceptual novelty.**
While the concrete LWA representation as a way of condition is new, the protocol of using a contidion to bridge privileged simulators to visual perception is well-discussed in recent research [1,2,3,4]. The core contribution is limited to the condition design of LWA itself, which serves mainly as a formal interface.

**Restricted validation domain.**
All main experiments center on autonomous-driving imagery and cross-domain demonstrations (robotics, indoor scenes) are qualitative only. The generalization claim of *universal hybrid world creation* lacks quantitative support.

**Evaluation focus on visual fidelity and controllability.**
Experiments show image quality and controllability metrics, but no metrics for physical consistency, temporal stability, or task-level realism are provided. The metrics being evaluated on do not imply benefits on real embodied agent training.

```
[1] Cosmos-Drive-Dreams: Scalable Synthetic Driving Data Generation with World Foundation Models.
[2] LucidSim: Learning Visual Parkour from Generated Images.
[3] SimGen: Simulator-conditioned Driving Scene Generation.
[4] NVIDIA Omniverse Blueprint: Synthetic Manipulation Motion Generation for Robotics.
```

**Questions:**

1. What are the exact digital form of the refined background layer? Are these arranged as pixels where each one of it is a semantic label from a predefined vocabulary? Or should they be a high-dimentional embedding, and if so, how is this supervised from the dataset perspective? Could you briefly explain the data structure of I/O for each stage in the pipeline?

2. For the extension to robotic manipulation, how is this LWA defined on indoor scenes? I suppose it still follows a object-layout-background design, but how are each layer defined specifically? Could you show me an example of how each item in Fig. 12 (b) correspond to layers?

3. Is there a particular reason why this pipeline cannot be verified with a real-world embodiment other than time-limitations for paper submission?

4. What is the philosophy behind LWA? Could you explain why the conditions needs to be designed in this way? If not structured as layers but other ways, I suppose the whole pipeline still works? I suppose we can still change background with instructional editing models?

---

> ### Author Response · Authors · 2025-11-21
>
> We thank the reviewer for the constructive feedback. We are encouraged that the reviewer found our Layered World Abstraction (LWA) design “new, interesting, and widely applicable across domains”, and our paper presentation to be “clear and detailed”. We address the concerns as follows.
>
> ### Weaknesses
>
> > W1 & Q3: Limited real-world verification
>
> We thank the reviewer for pointing out that our downstream evaluation on MetaVQA does not involve embodied agent training. This experiment demonstrates the improvements Dreamland brings to embodied scene understanding. In addition, we followed the conventional evaluation pipeline, including 3D perception, in SimGen and related works. We will rename Sec 5.4 and supp. D.5 in the camera-ready version. Due to hardware and regulation constraints, we didn't conduct real-world embodiment on autonomous driving.
>
> > W2: Clarification on novelty
>
> Indeed, our main contribution is introducing LWA as a flexible yet controllable bridge between simulators and generative models. While SimGen and LucidSim explore this direction, LucidSim relies on fully rendered scenes without hallucination, and SimGen sacrifices fine-grained control over simulator content. Thus, effectively bridging simulators and generative models for high-quality, controllable digital-twin/cousin data **remains an important but unsolved challenge**.
>
> We find that keeping Stage-1 condition maps precise and selectively edited is key to preserving grounded structure, which directly motivates introducing LWA to achieve this. As shown in Figure 8, LWA avoids hallucinating extra vehicles (left) and preserves correct pedestrian shapes (right). This controllability supports diverse applications, and Section 5.4 shows that Dreamland-generated data improves downstream detection and VLM performance.
>
> > W3: Restricted validation domain
>
> We appreciate the reviewer’s recognition of Dreamland’s broad applicability across multiple domains. While most of our quantitative experiments focus on the autonomous driving domain, we will revise to adopt a more measured tone on our claim regarding universal hybrid world creation in the camera-ready version.
>
> > W4: Evalution metrics
>
> We follow the conventional evaluation pipeline on fidelity and controllability used in prior works such as SimGen and MagicDrive. Thanks to the strong prior in pre-trained video generation models, together with our adaptive Stage-3 design, we preserve both physical consistency and temporal stability. The downstream tasks on 3D object detection and embodied scene understanding further demonstrate the benefits.
>
> ### Questions
>
> > Q1: LWA Structure and Module IO
>
> The refined background layer, same as other layers in LWA, is arranged with pixels containing depth and semantic labels. Stage-1 uses scene record to render different conditions that construct Sim-LWA. The Stage-2 receives Sim-LWA as input and outputs the transferred Real-LWA according to user instructions. The Real-LWA inputs into Stage-3 and outputs realistic scenes. For more details, please refer to Section 3.
>
> > Q2: LWA for manipulation tasks
>
> For robotic manipulation tasks, we follow the three-layer object-layout-background design with different semantic mapping. The object layer contains all the manipulatable objects, along with the robotic arm. The layout layer contains the operating surface and other physical barriers. The background layer covers the remaining scene. For Fig. 12 (b) left and middle, we preserve the object layer while making the background and layout layer editable. For Fig. 12 (b) right, we make the object layer editable to add objects.
>
> > Q4: Philosophy and unique design behind LWA
>
> We find that keeping Stage-1 condition maps precise and selectively edited is key to preserving grounded structure, which directly motivates introducing LWA to achieve this. Compared with SimGen, Dreamland’s LWA design provides stronger controllability of the condition maps, as shown in Figure 8: LWA avoids introducing additional vehicles (left) and preserves the correct pedestrian shape (right). Maintaining precise condition maps is critical for preserving the grounded structural information from Stage 1. Constructing the layered abstraction is also helpful for creating editing ground truth, and brings better flexibility at inference time, e.g., simulator-conditioned scene editing.

---

### Official Review · Reviewer_sLYN · 2025-10-30

**Soundness:** 3
**Presentation:** 3
**Contribution:** 3
**Rating:** 6
**Confidence:** 3

**Summary:**

This paper proposes Dreamland, a hybrid world generation framework that combines physics-based simulators with large-scale pretrained generative models for controllable scene creation. The key contribution is a Layered World Abstraction (LWA) that serves as an intermediate representation to bridge simulators and generative models. The approach involves three stages: (1) scene construction with a physics-based simulator, (2) Sim2Real transfer via instructional editing to align with real-world distributions, and (3) scene rendering with a large-scale pretrained generative model. The proposed pipeline is generalizable and scalable to unlock applications like video generation and scene editing. Additionally, the paper constructs a large-scale dataset called D3Sim for training and benchmarking hybrid generation pipelines combining simulators and generative models.

**Strengths:**

1. The proposed Layered World Abstraction (LWA) is a reasonable design enabling flexible control through preserved and editable regions, supporting diverse applications (scene editing, video generation, multiple simulators).
2. The proposed method achieves significant improvements compared to baseline (SimGen), 52.3% lower FID and 17.9% better si-RMSE. The experiments are comprehensive with quantitative metrics, user studies, ablations, and downstream task evaluations. The results in the demo video look fine.
3. D3Sim provides 60K paired samples with pixel-level alignment between simulation and real-world conditions for future research.

**Weaknesses:**

1. The technical novelty is limited. The pipeline primarily combines existing techniques (ACE++ for editing, standard diffusion adaptation). Innovation is mainly in integration and representation design rather than new methods.
2. Missing inference time comparisons with baselines. There are three stages, which might introduce additional processing time for generation.
3. Expensive Sim-Real paired data requirement. Stage-2 training requires costly pixel-aligned paired data. Unclear how to generalize to new simulators or domains without such data. Could we scale up the proposed method for more general usage, such as indoor scene generation?
4. Failure case analysis is absent. Where is the capability boundary of the proposed method? How could we achieve the best performance, and when will we fail?
5. For the multi-stage pipeline, the error accumulation problem and robustness were not analyzed.
6. The proposed method leverages a simulator to generate 3D scenes first as conditions for generative rendering, which is similar to previous 3D-conditioned generation methods, such as shape-for-motion [siggraph asia 2025], Image Sculpting [cvpr 2024], etc. And the idea of layered world representation is also widely used for scene generation, such LayerPano3D [siggraph 2025], HunyuanWorld 1.0, Scene4U [cvpr 2025], etc.

**Questions:**

Refer to weakness.

---

> ### Author Response · Authors · 2025-11-21
>
> We thank the reviewer for the constructive feedback. We are encouraged that the reviewer found our Layered World Abstraction (LWA) design “reasonable, flexible, and achieves improvements”, and our experiments to be “comprehensive”. We address the concerns as follows.
>
> ### Weaknesses
>
> > Q1. Clarification about novelty and innovation.
>
> Indeed, our main contribution is introducing LWA as a flexible yet controllable bridge between simulators and generative models. While SimGen and LucidSim explore this direction, LucidSim relies on fully rendered scenes without hallucination, and SimGen sacrifices fine-grained control over simulator content. Thus, effectively bridging simulators and generative models for high-quality, controllable digital-twin/cousin data **remains an important but unsolved challenge**.
>
> We find that keeping Stage-1 condition maps precise and selectively edited is key to preserving grounded structure, which directly motivates introducing LWA to achieve this. As shown in Figure 8, LWA avoids hallucinating extra vehicles (left) and preserves correct pedestrian shapes (right). This controllability supports diverse applications (as also noted by Reviewer sLYN), and Section 5.4 shows that Dreamland-generated data improves downstream detection and VLM performance.
>
> For Stage-2 and Stage-3, we follow common settings in the Generative AI community, highlight the generalizability of Dreamland and enable integration and simple adaptation to future, more potent pre-trained instructional editing and conditional generation models.
>
> > Q2. Inference time
>
> We report the inference time of Dreamland and baselines below. Dreamland achieves considerable generation speed with high resolution and frame numbers. Due to different resolutions and frame numbers, inference costs vary between methods for fair comparison.
> | Model           | Inference Time | Resolution-(Frames) |
> | --------------- | -------------- | ------------------- |
> | Dreamland       | 14s            | 1024x576            |
> | MagicDrive      | 9s             | 400x224             |
> | Panacea         | 20s            | 512x334-8           |
> | Dreamland-Video | 1400s          | 1280x704-121        |
>
> > Q3. Sim-real paired data
>
> Dreamland can indeed benefit from domain-specific sim–real paired data, and we have created D3Sim to support future training and research. However, Dreamland is designed to remain effective even when such paired data are difficult to collect at scale. As shown in Sec. D.3, we showcase how synthesized training data could be used for new domain. Thus, Dreamland has broader applicability beyond settings with readily available sim–real pairs.
>
> > Q4. Capability boundary and failure case
>
> Stage-1 simulator outputs are physically grounded and therefore rarely conflict with the Stage-3 pretrained generative model’s prior knowledge. In practice, Stage-3 handles corner cases well when provided with the condition maps, as demonstrated in the safety-critical scene in the supplementary video.
>
> *For the failure case,* we empirically find that the Dreamland-Video’s Stage-2 model is less robust than the Dreamland-Image’s Stage-2 model. We believe it is because one is initialized with editing model weights and one is initialized with text-conditioned generative model weights. We will add the analysis in the camera-ready version.
>
> > Q5. Error accumulation
>
> For error accumulation, our experiments on controllability (Table 1, 2, 7, and 9) calculate the condition alignment between pipeline input and output, which covers and accumulates the errors from different stages. Comparisons between Sim-LWA and Real-LWA in Table 6 and 7 further distinguish the error from each stage.
>
> > Q6. Related and prior work
>
> Actually, both our LWA design and scene-generation setup follow classical scene-parsing formulations [1,2], and we will update the Related Works section accordingly.
>
> [1] Layered Representation for Vision and Video, ICCV 1995
>
> [2] Scene Collaging: Analysis and Synthesis of Natural Images with Semantic Layers, ICCV 2013

---

### Comment · Area_Chair_AhWb · 2025-11-27

Dear Reviewers,

Thank you for your efforts in evaluating this submission. The current set of reviews shows a notable divergence in the overall scores. To ensure a fair and well-informed final decision, it is important that we have active participation from all reviewers during the author-reviewer discussion phase.

The authors have now responded to your comments. I kindly ask each of you to review their replies and engage in the discussion, especially to clarify whether their responses address your concerns and whether your initial assessment remains the same.

Your contributions at this stage are crucial for reaching a balanced consensus.
Thank you again for your time and commitment to the review process.

Best regards,

Area Chair

---

### Author Response · Authors · 2025-12-03
**Global Response**

We sincerely thank Area Chairs and the Reviewers for their time, careful evaluation, and constructive feedback. We especially appreciate the tremendous efforts of the newly assigned Area Chairs in carefully reviewing our rebuttal.

In the following overall summary, we provide a concise overview of the paper’s core contributions, summarize the reviewers’ feedback and main concerns, and highlight how each point was addressed and incorporated into the revised manuscript.

---

## ***Review Summary***

---

**Initial Reviews and Rebuttal Exchanges**: Our work, Dreamland, received *initial* ratings of  6 (`sLYN`), 4 (`5jdQ`), 4 (`m9s3`), and 6 (`FF93`). After we posted our detailed rebuttal responses, **reviewer `FF93` replied and raised the score to 8**. The remaining reviewers did not have time to respond to our rebuttal before the reviewer-response window closed.

**Key Strength**: The reviewers aligned on the key strength of our work:

- **The core design, Layered World Abstraction (LWA), is novel and effective.**
  - sLYN: “reasonable design”, “flexible”
  - 5jdQ: “Interesting abstraction design”
  - FF93: “novel hybrid formulation”

- **The pipeline is flexible and beneficial to downstream tasks.**
  - 5jdQ: “Wide applicability for other domains”
  - m9s3: “evaluation on downstream tasks to assess effectiveness”

- **The D3Sim dataset is useful and valuable.**
  - (all reviewers)

- **The paper is well-written.**
  - 5jdQ: “This paper is written in an easy-to-understand manner”
  - Presentation score: all **3** (good)

**Key Concerns and How We Addressed Them**: Alongside the positive feedback on Dreamland, reviewers raised several questions regarding additional ablation studies, theoretical insight, and clearer descriptions. We addressed each point in detailed individual responses accordingly. Below, we concisely summarize the main concerns and how we addressed them:

> Q1. Clarify the novelty. (raised by `sLYN` / `5jdQ` / `m9s3`)

While SimGen and LucidSim study how to bridge simulator and generative models, LucidSim relies on fully rendered scenes without hallucination, and SimGen sacrifices fine-grained control over simulator content. Thus, effectively bridging simulators and generative models for high-quality, controllable digital-twin/cousin data remains an important but unsolved challenge.

We find that keeping Stage-1 condition maps precise and selectively edited is key to preserving grounded structure, which directly motivates introducing LWA to achieve this. As shown in Figure 8, LWA avoids hallucinating extra vehicles (left) and preserves correct pedestrian shapes (right). This controllability supports diverse applications (as also noted by Reviewers `SLYN`, `5jdQ`, and `FF93`), and Section 5.4 shows that Dreamland-generated data improves downstream detection and VLM performance.

For Stage-2 and Stage-3, we follow common settings in the Generative AI community, highlight the generalizability of Dreamland and enable integration and simple adaptation to future, more potent pre-trained instructional editing and conditional generation models.

> Q2: Validation domain. (raised by `5jdQ` / `FF93`)

In Sec. 5.4 and App. D.3, we already demonstrated that Dreamland can be applied to indoor scenes beyond driving. While Dreamland has strong potential across other domains, we will further clarify this point and adopt a more measured tone on our claim regarding universal hybrid world creation in the camera-ready version.

---

We again thank the Area Chairs and Reviewers for their time and effort.

Best regards,

Authors of Submission #14652

---

### Meta-Review · Area_Chair_AzC1 · 2026-01-06

**Summary:**

This paper received mixed reviews, with two borderline accept (6) and two borderline reject (4) scores. The primary concerns raised by three reviewers relate to limited technical novelty and insufficient evidence of generalization beyond the evaluated domain. The authors have partially addressed the latter by conducting experiments in an additional domain (a table-setting task in the Isaac Sim robotics environment), suggesting that the proposed framework may be applicable beyond driving scenes.

However, I share the reviewers’ concerns regarding technical novelty. As noted in multiple reviews, the proposed method largely combines existing techniques, and the conceptual contribution relative to prior work remains limited. While I agree with the authors that the paper tackles an important and non-trivial problem that has not yet been fully resolved in the literature, I am not confident that the rebuttal and additional experiments would be sufficient to substantially change the reviewers’ overall assessments.

In addition, reviewer 5jdQ raised concerns about the lack of convincing real-world validation. Although real-world deployment, particularly for embodied agent or autonomous driving scenarios, is inherently challenging, the reviewers complained that the paper does not provide compelling evidence that the proposed model can lead to meaningful improvements in embodied agent performance or effectively translate to real-world tasks. Further empirical validation would be necessary to support such claims and to adequately address the reviewers’ concerns.

Overall, this was a difficult decision. Given the high level of competition at ICLR this year, I do not believe the paper meets the acceptance criteria this time. I encourage the authors to further address the reviewers’ comments and consider resubmission to a future venue.

**Reviewer Concerns:**

See the summary section.

**Reviewer Scores:**

See the summary section.

---

### Decision · Program_Chairs · 2026-01-26

Reject